# Composing Unbalanced Flows for Flexible Docking and Relaxation

**Gabriele Corso** [*]
MIT

**Vignesh Ram Somnath** [*]
ETH Zurich

**Noah Getz** [*]
MIT

**Regina Barzilay**
MIT

**Tommi Jaakkola**
MIT

**Andreas Krause**
ETH Zurich

## Abstract

Diffusion models have emerged as a successful approach for molecular docking, but they often cannot model protein flexibility or generate nonphysical poses. We argue that both these challenges can be tackled by framing the problem as a transport between distributions. Still, existing paradigms lack the flexibility to define effective maps between such complex distributions. To address this limitation, we propose *Unbalanced Flow Matching*, a generalization of Flow Matching (FM) that allows trading off sample efficiency with approximation accuracy and enables more accurate transport. Empirically, we apply Unbalanced FM on flexible docking and structure relaxation, demonstrating our ability to model protein flexibility and generate energetically favorable poses[1]. On the PDBBind docking benchmark, our method FlexDock improves the docking performance while increasing the proportion of energetically favorable poses from 30% to 73%.

## 1 Introduction

The mechanism of action of most drug molecules is often rationalized by studying how the molecule binds to one or more key proteins. Computationally, predicting the structure of these binding interactions constitutes the *molecular docking* task. Over the past decades, significant progress has been made in molecular docking, initially through classical search techniques and more recently with deep learning models. However, these methods primarily focus on rigid docking, assuming the protein has a fixed structure.

Proteins however are flexible entities and often undergo conformational changes during docking. Computational methods for understanding flexibility fall into two categories: *co-folding* and *flexible docking*. Co-folding involves predicting the bound structure of the protein and the ligand from scratch as a single task. Flexible docking, instead, aims at only modeling the limited structural transformation between unbound and bound protein structures, a formulation that allows for more efficient, interpretable, and controllable methods.

However, existing flexible docking methods have so far failed to provide satisfactory levels of accuracy. Among these, search-based techniques struggle to account for protein degrees of freedom due to the significantly increased dimensionality of the search space (Koes et al., 2013; McNutt et al., 2021). Deep learning methods have improved on this by extending diffusion or flow processes to include the protein flexibility, but these either severely limit the degrees-of-freedoms or implicitly force the model to learn protein folding entirely, resulting in structure predictions that are frequently inaccurate (Qiao et al., 2024; Lu et al., 2024). Furthermore, recent works have highlighted the tendency of ML docking models to generate non-physical structures (Buttenschoen et al., 2024).

Addressing the challenges above requires new techniques for precise transport between the complex distributions of unbound (apo) and bound (holo) protein structures. Flow matching (FM) (Lipman et al., 2022; Albergo et al., 2023), in principle, can learn such a transport. However, we show that

---

[*]Equal contribution. Correspondence to `gcorso@csail.mit.edu` and `vsomnath@inf.ethz.ch`.
[1]Our code and models are available at `https://github.com/vsomnath/flexdock`.

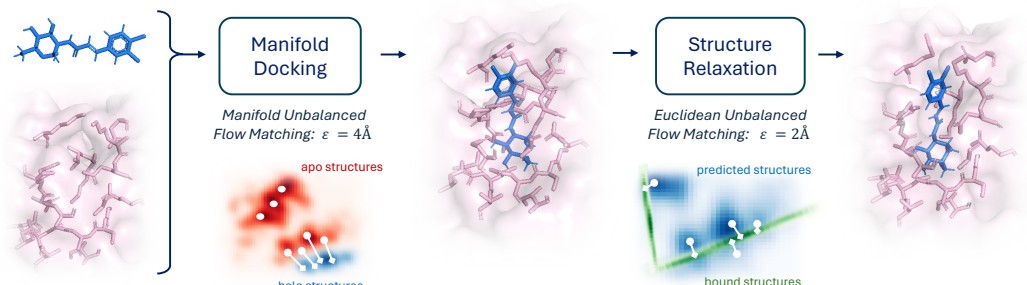

Figure 1: Schematic representation of the FLEXDOCK docking framework formed by the chaining of two unbalanced flows. The first (details in Section 4.1) is aimed at resolving the approximate pose by working on longer flows but reduced degrees of freedom, and the second (Section 4.2) aims at relaxing the generated structure producing high-quality poses.

directly applying flow matching to this task results in complex mappings, a difficult learning task, and consequently poor performance. To address this, we propose *Unbalanced Flow Matching*, where we relax the marginal constraints of FM through a more careful choice of coupling distribution. We study these choices both formally and empirically, resulting in a higher docking performance. We further consider composing short unbalanced flows and show that, in the limit, this corresponds to local likelihood gradient steps towards the desired marginal.

Based on the insights above, we construct a new flexible pocket-based docking method FLEXDOCK by chaining two separate Unbalanced Flows (see Figure 1). The first, referred to as *manifold docking*, is defined on a low dimensional space and aims to predict the approximate ligand pose and protein structural change. The second, *relaxation*, is defined on the full Euclidean space but with shorter maps and aims to provide a fast yet effective relaxation of the structure.

Empirically, we demonstrate that the new modeling perspective of FLEXDOCK enhances structure prediction quality, especially for protein conformations. On the PDBBind flexible pocket-based docking benchmark, FLEXDOCK improves the proportion of very accurate protein structure predictions (all-atom RMSD below 1Å) from 32% of DiffDock-Pocket to 42% while having the same training data and model architecture. Furthermore, thanks to the relaxation flow, the generated poses are far more physically realistic, as demonstrated by the proportion of poses passing PoseBusters checks (Buttenschoen et al., 2024) which increases from 30% to 73%.

In summary, we propose Unbalanced Flow Matching, a new distribution matching framework, and prove it provides a way to balance sample efficiency and sample quality. Furthermore, by chaining together new manifold docking and structure relaxation models based on Unbalanced FM, we develop a flexible docking method that addresses two major limitations of previous docking techniques: accurate protein flexibility modeling and the generation of energetically favorable poses.

## 2 BACKGROUND AND RELATED WORK

**Flow matching.** Given two distributions $q_0$ and $q_1$, Flow Matching (Lipman et al., 2022; Albergo et al., 2023) provides a way of learning a vector field $v_t$ which induces a continuous normalizing flow $\psi_t(\mathbf{x})$ that transports $q_0$ to $q_1$, i.e., $q_1(\mathbf{x}) = [\psi_1]_\# q_0(\mathbf{x})$, where $\#$ denotes the pushforward operator.

The key idea in FM is to define a conditional flow $\psi_t(\mathbf{x}_0|\mathbf{x}_1)$ interpolating between $\mathbf{x}_0 \sim q_0$ and $\mathbf{x}_1 \sim q_1$, and its associated vector field $u_t(\mathbf{x}_t|\mathbf{x}_1) = \frac{d}{dt}\psi_t(\mathbf{x}_t|\mathbf{x}_1)$. One can then learn the marginal vector field $v_t(\mathbf{x})$ with a neural network $v_t(\mathbf{x};\theta)$, by regressing against the conditional vector field with the conditional flow matching (CFM) objective:

$$\mathcal{L}_{\text{CFM}} = \mathbb{E}_{t,\mathbf{x}_0\sim q_0,\mathbf{x}_1\sim q_1} \|v_t(\mathbf{x}_t;\theta) - u_t(\mathbf{x}_t|\mathbf{x}_1)\|^2 \tag{1}$$

FM was further generalized by Pooladian et al. (2023) and Tong et al. (2023) who show that the CFM objective does not require independent samples from $q_0$ and $q_1$, but it can use an arbitrary joint sampling distribution $q(\mathbf{x}_0, \mathbf{x}_1)$, the coupling distribution, as long as it satisfies the marginal

constraints being $q_0$ and $q_1$ respectively. This formulation enables drawing a connection between FM and optimal transport (OT). When using OT to define the coupling distribution $q$, the flows become straight and the transport cost $\mathbb{E}_{q_0(\mathbf{x}_0)} \|\psi_1(\mathbf{x}_0) - \mathbf{x}_0\|^2$ is the OT cost $W_2^2(q_0, q_1)$. An extension of flow matching to the unbalanced setting was first explored in (Eyring et al., 2024), with couplings $q$ estimated using Unbalanced OT(UOT). In this work, we focus on arbitrary couplings $q$ and provide a formal and empirical characterization of the associated objectives.

**Flexible Docking.** Flexible docking assumes access to unbound structures of proteins (known as apo) and predicts how these will change upon ligand binding (producing holo-structures). Traditional *search-based* docking methods define a scoring function and search the space of possible poses, aiming to find the pose minimizing the scoring function (Alhossary et al., 2015; McNutt et al., 2021). These methods can in principle incorporate protein flexibility by adding torsion angles of the sidechains in the pocket to the search space. However, they struggle to find optimal joint poses due to the increased dimensionality and they are unable to model proteins' flexibility beyond the sidechains.

Recently, a few deep learning methods for flexible docking have been proposed. DIFFDOCK-POCKET (Plainer et al., 2023) and RE-DOCK (Huang et al., 2024) use diffusion and diffusion bridge models to capture the flexibility in protein pocket sidechains in addition to ligand flexibility, they, however, once again cannot model proteins' flexibility beyond sidechain torsion angles. DYNAMICBIND (Lu et al., 2024) incorporates backbone flexibility but uses hardcoded noise perturbation rules to interpolate from apo residue frames to holo residue frames. Somnath et al. (2023) explicitly predicts the conformational changes between apo and holo states of proteins using Diffusion Schrödinger Bridges by treating the apo and holo states as paired data.

**Structure Relaxation.** Even in rigid docking, ML methods, despite improving geometric prediction accuracy, often produce poses with nonphysical properties like steric clashes and strained bonds (Buttenschoen et al., 2024). These artifacts can hinder downstream prediction tasks, necessitating relaxation using molecular dynamics force fields. However, this relaxation process is often long (Bryant et al., 2024), complicating its use in large-scale screening.

## 3 UNBALANCED FLOW MATCHING

The motivation for flexible docking over co-folding is to leverage the unbound distribution of proteins and focus on modeling the precise effects of ligand binding. Formally, we wish to approximate a flow $\psi_t$ (equivalently vector field $v_t$), whose pushforward transports the distribution over apo-structures $q_0$ to the distribution over holo-structures $q_1$.

The flexible docking setting presents two challenges that make a direct application of flow matching for this task ineffective (as we will also demonstrate empirically). First, X-ray crystallography data are the main source of bound conformations, resulting in the availability of a single sample for holo-structure for most known complexes. This prevents one from using minibatch-OT based flow matching methods (Pooladian et al., 2023) and leads to a large expected length of conditional flows. Second, even if one could construct the optimal transport mapping $q$ between $q_0$ and $q_1$, the induced expected cost will likely remain very high. In fact, although the different conformational states of the protein do not change significantly upon ligand binding, their relative weights are often notably altered. Training FM in this setting requires the model to move the protein between conformations, leading to long and complex conditional and marginal flows, and a harder learning task.

We propose the Unbalanced Flow Matching framework to tackle such problems.

### 3.1 FORMALIZATION

Let $q_0$ be the input distribution, and $q_1$ a target distribution, defined over a data space $\mathcal{X}$. Unbalanced Flow Matching (UFM) aims to learn a continuous normalizing flow parameterized with a vector field $v_t(\mathbf{x}_t; \theta)$ that minimizes the following objective:

$$L_{\text{UFM}}(q, \theta) = \alpha \underbrace{\mathbb{E}_{t,(\mathbf{x}_0,\mathbf{x}_1) \sim q} \left[ \|v_t(\mathbf{x}_t; \theta) - u_t(\mathbf{x}_t | (\mathbf{x}_0, \mathbf{x}_1))\|^2 \right]}_{\text{Conditional flow matching}} + \underbrace{D_2(q_0 | q_{\mathbf{x}_0}) + D_2(q_{\mathbf{x}_1} | q_1)}_{\text{Mass-variation penalty}} \quad (2)$$

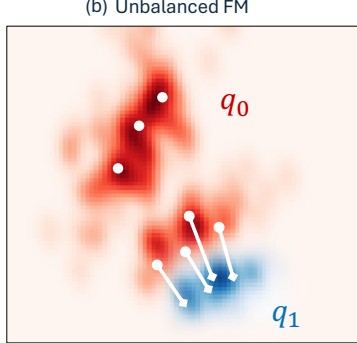

Figure 2: Schematic depiction of the difference between FM and Unbalanced FM. The first builds couplings that maintain the marginal distributions regardless of the difference between the distributions and the consequent expected transport cost. The second trades off some distributional coverage to obtain couplings with significantly reduced transport costs.

where $q$ is an arbitrary coupling distribution whose marginals we denote by $q_{\mathbf{x}_0}$ and $q_{\mathbf{x}_1}$, and $D_2$ is the Rényi Divergence of order 2. In Section 3.2, we show that this objective describes a tractable upper bound to the fundamental approximation vs efficiency trade-off.

Flow matching and its variants choose a coupling distribution $q$ such that marginals are conserved. Although this implies a zero *mass-variation penalty* in Equation 2, the *conditional flow matching* term can suffer significantly from this choice, as it might require the learning of complex couplings and consequently a poor approximation. In contrast, our framework jointly accounts for the flow matching and mass-variation penalty terms. However, jointly optimizing $q$ and $\theta$ to find a global minimum is typically intractable. Our approach thus relies on finding an approximation $q$, and using the flow matching objective with the chosen $q$. From Equation 2 (and assuming the space of $v_t(\cdot, \theta)$ includes the zero function), it is easy to see that:

$$\min_{q,\theta} L_{\mathrm{UFM}}(q,\theta) \leq \min_{q} \alpha \mathbb{E}_{t,(\mathbf{x}_0,\mathbf{x}_1)\sim q}\left[\|u_t(\mathbf{x}_t|(\mathbf{x}_0,\mathbf{x}_1))\|^2\right] + D_2(q_0|q_{\mathbf{x}_0}) + D_2(q_{\mathbf{x}_1}|q_1) \quad (3)$$

For $\mathcal{X} = \mathbb{R}^d$, $u_t(\mathbf{x}_t|(\mathbf{x}_0,\mathbf{x}_1)) = \mathbf{x}_1 - \mathbf{x}_0$ is a common choice. Plugging this into Equation 3 gives us:

$$\min_{q,\theta} L_{\mathrm{UFM}}(q,\theta) \leq \min_{q} \mathbb{E}_{t,(\mathbf{x}_0,\mathbf{x}_1)\sim q}\left[\alpha\|\mathbf{x}_1 - \mathbf{x}_0\|^2\right] + D_2(q_0|q_{\mathbf{x}_0}) + D_2(q_{\mathbf{x}_1}|q_1) \triangleq \mathrm{UOT}(q_0,q_1) \quad (4)$$

The upper bound corresponds to the Unbalanced Optimal Transport (UOT) objective with quadratic cost $\alpha\|\mathbf{x}_1 - \mathbf{x}_0\|^2$ that we can approximate to choose $q$. Fixing $q$, we can then learn $\theta$ via SGD with the resulting simplification of Equation 2:

$$\mathcal{L}_{\mathrm{UFM}}(\theta;q) = \mathbb{E}_{t,(\mathbf{x}_0,\mathbf{x}_1)\sim q}\left[\|v_t(\mathbf{x}_t;\theta) - u_t(\mathbf{x}_t|(\mathbf{x}_0,\mathbf{x}_1))\|^2\right] \quad (5)$$

By lifting the mass-conservation requirement of FM, we now simplify the learning task (see Figure 5 for an empirical validation). However, now even with a perfectly trained vector field, the pushforward of $q_0$ will no longer correspond to $q_1$. Our goal of obtaining samples from $q_1$, therefore, involves reweighting the samples generated by the learned flow. In practice, we approximate this re-weighting with a discriminator $\mathbf{c}$ akin to the confidence model used in (Corso et al., 2022) (see Algorithm 1, and Section 4.1).

---

**Algorithm 1:** UNBALANCED FM INFERENCE

---

Sample prior $\mathbf{x_0}^{(i)} \sim q_0$

Use flow $\hat{\psi}_1(\cdot|q,\theta)$ to map $\mathbf{x_0}^{(i)}$ to $\mathbf{x_1}^{(i)}$

Reweight or reject with learned discriminator $\mathbf{c}$

Return $[\mathbf{x_1}^{(1)}, \mathbf{x_1}^{(2)}, ..., \mathbf{x_1}^{(n)}]$

---

### 3.2 Efficiency vs Approximation Trade-off

In this section, we show how the UFM objective arises from optimizing the trade-off between sample efficiency and approximation accuracy. Sample efficiency refers to the number of samples from $q_0$ needed to obtain an approximate sample of $q_1$ (passing the discriminator) under the learned flow. These approximate samples of $q_1$ still incur some imprecision due to the flow not being learned perfectly; the approximation accuracy refers to the Wasserstein distance between $q_1$ and the distribution of these approximate samples. To study this, we consider a modified version of the Unbalanced FM inference procedure (Algorithm 1). The main difference of this procedure (Algorithm 2) is that we perform some filtering not only after the flow but also before it. Algorithm 2 is, thus, at most as sample efficient as Algorithm 1 but easier to study.

---

**Algorithm 2:** UFM Efficiency Lower Bound

Sample prior $\mathbf{x_0}^{(i)} \sim q_0$

Reweight or reject based on approx. $\frac{q_{\mathbf{x_0}}(\mathbf{x_0}^{(i)})}{q_0(\mathbf{x_0}^{(i)})}$

Use flow $\hat{\psi}_1(\cdot|q, \theta)$ to map $\mathbf{x_0}^{(i)}$ to $\mathbf{x_1}^{(i)} \sim \hat{q}_{\mathbf{x_1}}$

Reweight or reject based on approx. $\frac{q_1(\mathbf{x_1}^{(i)})}{q_{\mathbf{x_1}}(\mathbf{x_1}^{(i)})}$

Return $[\mathbf{x_1}^{(1)}, \mathbf{x_1}^{(2)}, ..., \mathbf{x_1}^{(n)}]$

---

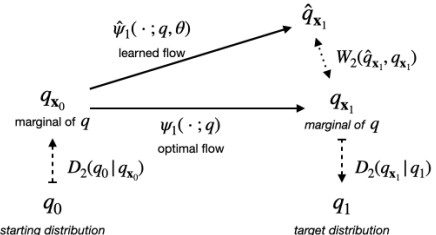

Algorithm 2: High-level inference routine for modified Unbalanced FM routine used to define theoretical a lower bound on the efficiency of the original Unbalanced FM routine of Algorithm 1.

Figure 3: Relationship between the different distributions introduced in the theoretical analysis of the Unbalanced FM efficiency lower bound algorithm.

**Formalization.** Let $\psi_1(\cdot; q)$ be the optimal flow from the Unbalanced FM objective with couplings $q$ and $\hat{\psi}_1(\cdot; q, \theta)$ its approximation we are able to learn. FM guarantees us that $q_{\mathbf{x_1}} = [\psi_1(\cdot|q)]_{\#} q_{\mathbf{x_0}}$ and let $\hat{q}_{\mathbf{x_1}}(\cdot|\theta) = [\hat{\psi}_1(\cdot|q, \theta)]_{\#} q_{\mathbf{x_0}}$. We summarize all the defined distributions and their relationship in Figure 3.

The definition of Unbalanced FM as a method to bridge two distributions $q_0$ and $q_1$ leads us to analyze the trade-off between the approximation error when learning the flow, formalized as $W_2^2(\hat{q}_{\mathbf{x_1}}(\cdot|\theta), q_{\mathbf{x_1}})$, and the sample efficiency $\text{ESS}^*(q)$ that derives from having to perform rejection sampling to bridge the gaps between $q_0$ and $q_{\mathbf{x_0}}$ and between $q_{\mathbf{x_1}}$ and $q_1$. Simple mappings will result in low approximation errors but potentially lower efficiency, and vice versa.

**Approximation error.** Using Theorem 1 from Benton et al. (2023) we can show, under appropriate assumptions that the approximation error for a given coupling distribution $q$, $W_2^2(\hat{q}_{\mathbf{x_1}}(\cdot|\theta), q_{\mathbf{x_1}})$, is bounded by FM objective:

$$W_2^2(\hat{q}_{\mathbf{x_1}}(\cdot|\theta), q_{\mathbf{x_1}}) \leq L^2 \cdot \mathbb{E}_{t,q}\left[\|v_t(\mathbf{x}_t; \theta) - v_t(\mathbf{x}_t)\|^2\right] \leq L^2 \cdot \mathbb{E}_{t,q}\left[\|v_t(\mathbf{x}_t; \theta) - u_t(\mathbf{x}_t|\mathbf{x}_1)\|^2\right] \quad (6)$$

where $L = \exp[\int_0^1 L_t dt]$ and $L_t$ is a constant such that $v_\theta(\mathbf{x}, t)$ is $L_t$-Lipschitz in $\mathbf{x}$. The second inequality follows from the convexity of the squared norm.

**Sample efficiency.** We can measure the sample efficiency that the model has when converting samples of $q_0$ to unbiased samples of $q_1$ via the effective sample size $\text{ESS}^*(q)$, i.e., the reciprocal of how many samples from $q_0$ it takes using the ideal flow $\psi_1(\cdot, q)$ to generate an unbiased sample from $q_1$. In Proposition 1 (derivation in Appendix A.1), we demonstrate that for an ideal flow this sample efficiency is bounded by the similarity between $q_0$ and $q_1$ and the respective marginals of $q$:

**Proposition 1.** *The effective sample size, ESS\*, for sampling $q_1$ when having access to samples of $q_0$ and a perfectly trained flow with coupling distribution $q$ is bounded by:*

$$ESS^*(q) \geq \exp\left[-D_2(q_0|q_{\mathbf{x_0}}) - D_2(q_{\mathbf{x_1}}|q_1)\right]. \quad (7)$$

**Trade-off upper bound.** The trade-off between approximation error and sampling efficiency in the choice of optimal flow can then be expressed as a joint objective:

$$\beta \underbrace{W_2^2(\hat{q}_{\mathbf{x}_1}(\cdot|\theta), q_{\mathbf{x}_1})}_{\text{Approximation error}} - \underbrace{\log \text{ESS}^*(q)}_{\text{Sampling efficiency}} \leq L_{\text{UFM}}(\theta, q) \tag{8}$$

where the inequality derives from setting $\alpha = L^2 \beta$ and using the derivation above. This motivates the UFM objective as a way to optimize an upper bound on the joint approximation error and sample (in-) efficiency trade-off.

In practice, since we only have access to one sample for the distribution over bound structures (the crystal structure in PDB, typically unique), we cannot define $q$ via exact optimal Unbalanced OT. In our experiments, we approximate the optimal coupling with the distribution $q(\mathbf{x}_0, \mathbf{x}_1) \propto q_0(\mathbf{x}_0)q_1(\mathbf{x}_1)\mathbb{I}_{c(\mathbf{x}_0,\mathbf{x}_1)<c_{\text{task}}}$, where $c(\mathbf{x}_0, \mathbf{x}_1)$ is some, task-specific, transport cost, and $c_{\text{task}}$ is an empirically chosen cutoff to balance sample efficiency and mapping complexity.

### 3.3 COMPOSING UNBALANCED FLOWS

In Section 4.2, we will show how we obtain effective structure relaxations by learning a further unbalanced flow with smaller matching costs on top of the distribution learned by the initial manifold docking unbalanced flow. In this section, we analyze the effect of composing flows together in the case of short distance couplings that corresponds to the setting of structure relaxation.

Each flow tries to learn the matching from the output distribution of the previous flow and the final distribution. We consider the specific coupling introduced in the previous section $q(\mathbf{x}_0, \mathbf{x}_1) = q_0(\mathbf{x}_0)q_1(\mathbf{x}_1)\mathbb{I}_{\|\mathbf{x}_0-\mathbf{x}_1\|\leq\epsilon}$ with $q_0$ now representing the output distribution of the previous flow. In Proposition 2 (derivation in Appendix A.2), we demonstrate that taking $\epsilon \to 0$ leads to approximate local likelihood gradient steps that push $q_0$ towards matching $q_1$:

**Proposition 2.** *A coupling* $q(\mathbf{x}_0, \mathbf{x}_1) \propto q_0(\mathbf{x}_0)q_1(\mathbf{x}_1)\mathbb{I}_{\|\mathbf{x}_0-\mathbf{x}_1\|\leq\epsilon}$ *with* $\epsilon \to 0$ *generated a marginal flow* $\psi_1$ *that has the form:*

$$\psi_1(\mathbf{x}_0) \approx \mathbf{x}_0 + C_{\epsilon,n}[q_0(\mathbf{x}_0)\nabla q_1(\mathbf{x}_0) - q_1(\mathbf{x}_0)\nabla q_0(\mathbf{x}_0)] \tag{9}$$

*where* $C_{\epsilon,n}$ *is a constant that depends on* $\epsilon$ *and the dimensionality of the space* $n$.

therefore, for all the points $\mathbf{x}_0$ sampled from $q_0$ that have low likelihood under the final distribution $q_1$ (so $q_1(\mathbf{x}_0) << q_0(\mathbf{x}_0)$), the transform $\psi_1(\mathbf{x}_0) \approx \mathbf{x}_0 + C_{\epsilon,n}q_0(\mathbf{x}_0)\nabla_{\mathbf{x}_0}q_1(\mathbf{x}_0)$ will push the sample toward the regions of higher likelihood of $q_1$. Notably, the learned flows also have the property $\psi_1(\mathbf{x}_0) = \mathbf{x}_0$ when $q_0 = q_1$.

## 4 FLEXIBLE DOCKING

In the flexible docking task, our goal is to learn the joint distribution over the bound structures (equivalently, poses thereof) of a protein-ligand complex given the distribution over the unbound structures of the protein. We focus on the pocket-based flexible docking task, but all components of our framework can be applied to the blind docking setting as well (unknown protein pocket).

In the previous section, we motivated the necessity of Unbalanced FM for the flexible docking task, and the attractive properties that emerge from composing unbalanced flows. In this section, we leverage these insights to decompose the flexible docking task into two subtasks, both modeled as unbalanced flows: i) docking over the manifold degrees of freedom (Section 4.1), and ii) structure relaxation (Section 4.2).

### 4.1 DOCKING OVER MANIFOLD DEGREES OF FREEDOM

Ligand and protein poses can be regarded as elements of $\mathbb{R}^{3n_l}$ and $\mathbb{R}^{3n_p}$, where $n_l$ and $n_p$ are the number of atoms in the ligand and protein respectively. However, during docking, ligand flexibility is largely concentrated in the torsion angles at rotatable bonds (Jing et al., 2022), while for proteins, the flexibility lies in the backbone frames and sidechain torsion angles (Jumper et al., 2021). Motivated by the success of Intrinsic Diffusion Models (Corso, 2023) in similar domains, we reduce the space of ligand and protein poses by defining our generative model over these degrees of freedom.

**Transport of distributions formulation.** For the distribution over ligand poses, we largely follow DIFFDOCK (Corso et al., 2022), learning a diffusion model over the product space of rotations, translations, and torsions, $\mathbb{P} = SO(3) \times \mathbb{R}^3 \times \mathbb{T}^{m_l}$. To model conformational changes in protein structures upon docking, we employ our Unbalanced FM framework. The prior $q_0$ is defined as the distribution of (computationally generated) unbound structures, while the target distribution $q_1$ is defined over the crystallized bound structures. For the prior, given the current inability of computational models to sample the diverse conformational space (Jing et al., 2024), we use outputs from ESMFold (Lin et al., 2022) with the addition of small Gaussian noise. Designing an Unbalanced FM objective then amounts to choosing a coupling $q(\mathbf{x}_0, \mathbf{x}_1)$, a conditional probability path $p_t(\mathbf{x}|\mathbf{x}_0, \mathbf{x}_1), (\mathbf{x}_0, \mathbf{x}_1) \sim q$, and the associated conditional vector field $\mathbf{u}_t(\mathbf{x}|\mathbf{x}_0, \mathbf{x}_1)$.

**Choice of coupling** $q$. A key requirement for $q$ is to be able to sample pairs during training. Because we typically only have access to one sample for the distribution over bound structures (the crystal structure in PDB, typically unique), we cannot define $q$ via Unbalanced OT. Therefore, we approximate the optimal coupling with the distribution $q(\mathbf{x}_0, \mathbf{x}_1) \propto q_0(\mathbf{x}_0)q_1(\mathbf{x}_1)\mathbb{I}_{c(\mathbf{x}_0, \mathbf{x}_1) < c_{\text{dock}}}$, where $c(\mathbf{x}_0, \mathbf{x}_1)$ is defined as the aligned RMSD between the C$\alpha$ positions of the residues in the pocket and its neighborhood, and $c_{\text{dock}}$ is an empirically chosen cutoff to balance sample efficiency and mapping complexity. We can sample from $q$ by taking independent samples from $q_0$ and $q_1$ and rejecting if $c(\mathbf{x}_0, \mathbf{x}_1) \geq c_{\text{dock}}$.

**Flow Matching on** $SE(3)$ **and** $\mathbb{T}$. For a protein with $n$ residues and $m_p$ sidechain torsions, we define the flow over the product space $SE(3)^n \times SO(2)^{m_p}$, where the $SE(3)$ frame for each residue corresponds to a roto-translation around the C$\alpha$ atom, and the hypertorus $\mathbb{T}^{m_p}$ over sidechain torsions.

Following the disintegration of measures (Pollard, 2002), every $SE(3)$-invariant measure can be broken down into a $SO(3)$-invariant measure and a measure proportional to the Lebesgue measure on $\mathbb{R}^3$, allowing us to build flows independently on $SO(3)$ and $\mathbb{R}^3$. Following Chen & Lipman (2024), given two points $(\mathbf{x}_0, \mathbf{x}_1) \sim q$, the conditional probability path between $\mathbf{x}_0$ and $\mathbf{x}_1$ is given by the geodesic between them, $\mathbf{x} = \exp_{\mathbf{x}_0}(t \log_{\mathbf{x}_0}(\mathbf{x}_1))$, and the corresponding conditional flow is $\mathbf{u}_t(\mathbf{x}_t|\mathbf{x}_0, \mathbf{x}_1) = \frac{\log_{\mathbf{x}_t} \mathbf{x}_1}{1-t}$.

For $SO(3)$, the geodesics can be computed efficiently by using the axis-angle representation (equivalent to $\log(\mathbf{x}_1)$) and the parallel transport operation (left multiplication with $\mathbf{x}_0$), while exp is simply the matrix exponential. We view the torus $\mathbb{T}$ as the quotient space $\mathbb{R}/2\pi\mathbb{Z}$, thus $\exp_{\mathbf{x}_0}(\mathbf{x}_1) = (\mathbf{x}_0 + \mathbf{x}_1) \mod 2\pi$ (equivalent to wrapping around $\mathbb{R}$), and $\log_{\mathbf{x}_0} \mathbf{x}_1 = \arctan 2(\sin(\mathbf{x}_1 - \mathbf{x}_0), \cos(\mathbf{x}_1 - \mathbf{x}_0))$.

**Confidence discriminator.** As discussed in Section 3.1, it is necessary to apply a rejection or filtering step at the end of the unbalanced flow. This is because, by the design of the Unbalanced FM framework, some of the samples produced will be incorrect (i.e. far from high-likelihood regions of $q_1$). To do this we train a confidence model (Corso et al., 2022) to predict the likelihood that a sample from $q_1$ is within 2Å ligand-RMSD and 1Å AA-RMSD of the input pose. This confidence score is then used to rank or select different samples obtained from the flow.

## 4.2 STRUCTURE RELAXATION

Motivated by the results from Section 3.3, we frame structure relaxation as an (unbalanced) flow, where the prior $q_0$ is now the distribution over poses generated by the manifold docking process in Section 4.1, and the target distribution $q_1$ is still defined over the crystallized bound structures.

**Choice of coupling** $q$. We adopt a similar definition for $q$ as Section 4.1, except now $c(\mathbf{x}_0, \mathbf{x}_1)$ is defined as the average of the aligned atomic RMSDs between the predicted pose for both the ligand and protein (not just the latter) and their ground truth counterparts. We also use a smaller cutoff $c_{\text{relax}} < c_{\text{dock}}$ to enforce small structural changes.

**Flow Matching for Relaxation.** We define the flow over the Euclidean space. Given $(\mathbf{x}_0, \mathbf{x}_1) \sim q$, the conditional probability path $p_t(\mathbf{x}|\mathbf{x}_0, \mathbf{x}_1)$ is defined as $\mathbf{x}_0 \cdot (1 - t) + \mathbf{x}_1 \cdot t$, and the associated conditional vector field $\mathbf{u}_t(\mathbf{x}|\mathbf{x}_0, \mathbf{x}_1) = \frac{\mathbf{x}_1 - \mathbf{x}}{1-t}$.

**Energy-based loss.** Molecular structures are characterized by degrees of freedom with vastly different stiffness, for example the Generalized Amber Force Field (GAFF) is specified such that a single bond between Carbon and Nitrogen atoms allows oscillations with a vibrational frequency of $4401 \text{ cm}^{-1}$ (Wang et al., 2004). This causes even small $W_2$ approximation errors to be particularly problematic along such degrees of freedom. Therefore, to further improve the flow's ability to accurately relax degrees of freedom with narrow distributions, we introduce an additional energy-based loss to encourage the model to sample low-energy poses.

A similar approach has been explored in the scope of works on Boltzmann Generators (Noé et al., 2019) where models are trained directly to minimize the reverse KL divergence between the generated samples and some energy function by evaluating the energy of the generated poses and their log-likelihood under the model. Our approach distinguishes from this in two main aspects: firstly, to preserve the stable and efficient training regime of (unbalanced) flow matching we do not use full sampling trajectories but only evaluate the energy of the $\hat{\mathbf{x}}_1 = \mathbf{x}_t + (1-t)v_t(\mathbf{x}_t; \theta)$ predicted poses for $t$ close to 1; secondly, instead of using force-field based energy functions that tend to be very unstable on non-equilibrium poses we employ a handcrafted flat-bottom potential with the following form:

$$\mathcal{L}_{\text{energy}} = \sum_{i,j} \max\left(\left\|\hat{\mathbf{x}}_1^{(i)} - \hat{\mathbf{x}}_1^{(j)}\right\| - U_{i,j}, 0\right) + \max\left(L_{i,j} - \left\|\hat{\mathbf{x}}_1^{(i)} - \hat{\mathbf{x}}_1^{(j)}\right\|, 0\right) \qquad (10)$$

where $L_{i,j}$ and $U_{i,j}$ are lower and upper bounds (if any) on the distance between each pair of atoms $(i, j)$ and depend on whether atoms $i$ and $j$ are covalently bonded, substituents of the same central atom, or constitute an intermolecular cross term between the protein and ligand. Further details on the formulation are provided in Appendix B.1.

**Energy filtering.** During inference, we utilize this flat-bottom potential combined with the initial poses confidence scores as the Unbalanced FM discriminator to select the predictions for which relaxation was successful. Starting with conformers predicted by the flexible docking model, we use the docking confidence model to score and prioritize the predictions. After relaxation, we use the flat bottomed potential to select only the relaxed conformers for which the potential is zero. We then choose the relaxed pose with the greatest confidence score. If all relaxed conformers have a non-zero potential value, we instead return the highest confidence conformer prior to relaxation.

## 5 Experiments

**Benchmark.** We train and test our models on the widely adopted PDBBind benchmark (Liu et al., 2017). We use computationally generated structures from ESMFold (Lin et al., 2022) as samples from the distribution of unbound structures. We also evaluate on PoseBusters, a recent benchmark dataset curated from the PDB, with several filtering steps and sequence-based clustering. Since we focus on the flexible docking task, we evaluate the accuracy of both the predicted ligand and pocket atom poses. We first align the predicted pocket atoms to their ground truth counterparts, and then compute the heavy-atom RMSD between the predicted and ground truth poses for the ligand (with permutation symmetry correction) and pocket atoms. For the ligand, following Huang et al. (2024), we report the median and the percentage of predictions with RMSD below 2Å. For the pocket atoms, we similarly report the percentage of predictions with RMSD below 1Å. Finally, we report the pose quality using the percentage of top-1 predictions passing the series of tests from Buttenschoen et al. (2024) referred to as PoseBusters checks.

**Baselines.** For the PDBBind benchmark, we compare FLEXDOCK to well established search-based methods SMINA and GNINA, and flexible ML-based pocket level docking methods in DIFFDOCK-POCKET (Plainer et al., 2023) and RE-DOCK (Huang et al., 2024). All methods receive as input – i) the ligand with an initial seed conformation (using RDKit), ii) the ESMFold predicted structure, and iii) the pocket residues. Additional details regarding the experimental setup, data, and baselines can be found in Appendix D along with a link to the code used for our experiments. For the PoseBusters benchmark we also report other baselines for pocket-based docking – these include both rigid methods taking in the holo-structure like DEEPDOCK (Liao et al., 2019), UNIMOL (Zhou et al., 2023), GOLD (Verdonk et al., 2003) and VINA (Trott & Olson, 2010) as well as ML-based cofolding methods UMOL (Bryant et al., 2024) and ALPHAFOLD3 (Abramson et al., 2024).

Table 1: **Top-1 PDBBind ESMFold Docking Performance** We standardize all the generative modeling-based methods to take 10 samples. - indicates results missing due to baselines taken from previous work where output structures were not provided. *Note that REDOCK uses a slightly different definition of pocket making the results not fully comparable, while their code is not available.

| Method | Ligand RMSD | | All-Atom RMSD | % PB | % PB valid and | Runtime (s) |
| | % < 2Å ↑ | Median Å ↓ | % < 1Å ↑ | valid ↑ | L-RMSD < 2 Å ↑ | |
|---|---|---|---|---|---|---|
| SMINA (rigid) | 6.6 | 7.7 | N.A. | - | - | 258 |
| SMINA | 3.6 | 7.3 | 5.2 | - | - | 1914 |
| GNINA (rigid) | 6.7 | 7.1 | N.A. | **93.3** | 6.7 | 260 |
| GNINA | 8.4 | 7.9 | 4.5 | 91.3 | 7.7 | 1575 |
| DIFFDOCK-POCKET (rigid) | 37.5 | 3.0 | N.A. | 25.9 | 4.9 | 17 |
| DIFFDOCK-POCKET | **41.8** | **2.5** | 32.4 | 30.1 | 5.9 | 17 |
| REDOCK* | 39.0 | **2.5** | 39.8 | - | - | 15 |
| FLEXDOCK | 39.7 | **2.5** | **41.7** | 72.9 | **33.7** | **11** |

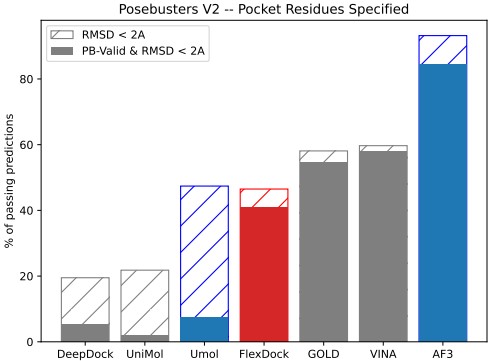

Figure 4: Performance of different pocket-based methods on the PoseBusters v2 benchmark. GOLD, VINA, DeepDock and UniMol take as input the holo-structure of the protein, while UMol, FlexDock and AlphaFold3 only the sequence. AlphaFold3 was trained on a dataset ∼10x larger.

Figure 5: Docking performance for different cutoffs $c_{dock}$. FM corresponds to UFM with $c_{dock} = \infty$. We generate 10 samples using a small model (4M parameters) without any additional filtering or relaxation, and evaluate the best prediction of these samples for every complex.

**Results.** Table 1 presents the comparison of previous methods in the field with the overall FLEXDOCK model which shows improvements in many metrics. To better understand the value of our methodological contributions, it is useful to directly compare with DIFFDOCK-POCKET, as this uses the same architecture and training regime. Here we see large improvements in the pocket all-atom accuracy with AA-RMSD < 1 Å improving from 32% to 42% and the pose quality (measured by the proportion of PB valid poses) improving from 30% to 73%.

On PoseBusters, our method performs significantly better than the fast ML-based rigid docking models DEEPDOCK and UNIMOL even without receiving the holo-structure. When compared to large-scale co-folding models, it has similar RMSD performance but significantly better validity than UMOL while being significantly faster (average of 11s per complex for FLEXDOCK and 206s for UMOL excluding its MSA computation step). ALPHAFOLD3 performs significantly better, however, it has been trained on more than one order of magnitude extra data compared to FLEXDOCK and UMOL.

**Ablations.** Ablations regarding the choice of cutoff $c_{dock}$ (See Para. 4.1) can be found in Figure 5. The comparison with standard FM highlights the importance of unbalanced flows in tackling this task. Using a cutoff of 4Å gives better performance across both ligand and protein pose prediction metrics.

We also perform ablation studies in Table 2 to decompose the improvements from the different components of FLEXDOCK. This shows that, on the one hand, the improvement in All-Atom RMSD

Table 2: Ablation of the effect of the various components of the relaxation unbalanced flow described in Section 4.2.We now split the runtime into the docking time + relaxation time.

| Method | Ligand RMSD | | All-Atom RMSD | % PB | % PB valid and | Runtime (s) |
| | % < 2Å ↑ | Median Å ↓ | % < 1Å ↑ | valid | L-RMSD < 2 Å | |
|---|---|---|---|---|---|---|
| FLEXDOCK (no relaxation) | 38.8 | 2.6 | 43.8 | 13.1 | 9.4 | 10 |
| FLEXDOCK (OpenMM Relaxation) | 38.2 | 2.6 | 42.1 | 68.4 | 30.3 | 10 + 155 |
| FLEXDOCK (relaxation w/out energy loss) | 40.9 | 2.5 | 44.1 | 17.3 | 12.1 | 10 + 0.3 |
| FLEXDOCK (no energy filtering) | 40.4 | 2.5 | 40.6 | 64.5 | 32.2 | 10 + 0.3 |
| FLEXDOCK | 39.7 | 2.5 | 41.7 | 72.9 | 33.7 | 10 + 0.3 |

can be attributed to the improved modeling of protein flexibility through the Unbalanced FM over the manifold degrees of freedom, not the relaxation. On the other hand, the improvement in pose quality can be largely attributed to the different steps of the relaxation Unbalanced FM which improves the %-PB valid from 13 to 73. Among these, we highlight the critical role of the energy loss. We additionally compare our relaxation to the OpenMM relaxation offered in the PoseBusters benchmark. Our relaxation routine achieves an improvement of 4.5% in PB-Valid metrics, while being at least 2 order of magnitude faster.

## 6 CONCLUSION

While generative models are revolutionizing the field of structural biology, existing frameworks lack the flexibility needed to tackle outstanding domain-specific issues. In this work, we presented Unbalanced Flow Matching, a generalization of the Flow Matching framework that relaxes the marginal preservation assumption. Theoretically, we showed this approach allows us to improve the approximation error FM incurs at the cost of some sample efficiency. We apply Unbalanced FM to enhance the ability of flexible docking models to model the flexibility of protein structures and predict poses with good energetic properties. Empirically, we show that the resulting framework FLEXDOCK improves these properties over the best existing methods on the PDBBind benchmark.

## ACKNOWLEDGEMENTS

We thank Ya-Ping Hsieh, Itamar Chinn, Hannes Stark, and Bowen Jing for their valuable feedback and insightful discussions.

This work was supported by NCCR Catalysis (grant numbers 180544 and 225147), a National Centre of Competence in Research funded by the Swiss National Science Foundation, the NSF Expeditions grant (award 1918839: Collaborative Research: Understanding the World Through Code), the Abdul Latif Jameel Clinic for Machine Learning in Health, the DTRA Discovery of Medical Countermeasures Against New and Emerging (DOMANE) Threats program, the MATCHMAKERS project supported by the Cancer Grand Challenges partnership financed by CRUK (CGCATF-2023/100001) and the National Cancer Institute (OT2CA297463) and the Machine Learning for Pharmaceutical Discovery and Synthesis (MLPDS) consortium.

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

# A PROOFS

Note: in all derivations and definitions in this section, we will assume that the distributions we work with are defined in Euclidean space and have full support.

## A.1 UNBALANCED FM SAMPLE EFFICIENCY

**Proposition 1.** *The effective sample size, ESS\*, for sampling $q_1$ when having access to samples of $q_0$ and a perfectly trained flow with coupling distribution q is bounded by:*

$$\text{ESS}^*(q) \geq \exp\left[-D_2(q_0|q_{x_0}) - D_2(q_{x_1}|q_1)\right] \tag{11}$$

*where $D_2$ is the Rényi Divergence of order 2.*

*Proof.* When comparing pairs of distributions $p$ and $q$ the (population) effective sample size $\text{ESS}^*(p, q)$ is defined as (Maia Polo & Vicente, 2023):

$$\text{ESS}^*(p, q) := \exp[-D_2(p|q)]$$

and it can be considered as the percentage of effective samples from $q$ one can obtain when taking samples from $p$.

Similarly, we define $\text{ESS}^*(q)$ in our setting as the percentage of effective samples from $q_1$ one can obtain from $q_0$ using $\psi_1$. While $\psi_1$ could be directly applied to any distribution, including $q_0$, it is hard to model its pushforward analytically. On the other hand, we know that $q_{x_1}$ is the pushforward of $q_{x_0}$, therefore we can obtain samples from $q_1$ by (1) reweighting samples of $q_0$ into samples of $q_{x_0}$, (2) transporting samples from $q_{x_0}$ to samples of $q_{x_1}$ and (3) reweighting samples from $q_{x_1}$ into samples from $q_1$. By assumption of perfect flow step (2) has perfect efficiency, however, steps (1) and (3) both may require more than one sample in expectation to be unbiased. This translates into an efficiency equal to the product of the two effective sample sizes:

$$\text{ESS}^*(q) \geq \text{ESS}^*(q_0, q_{x_0}) \text{ ESS}^*(q_{x_1}, q_1) = \exp\left[-D_2(q_0|q_{x_0})\right] \exp\left[-D_2(q_{x_1}|q_1)\right]$$

$$= \exp\left[-D_2(q_0|q_{x_0}) - D_2(q_{x_1}|q_1)\right].$$

where the inequality derives from the possibility of the existence of more effective procedures for this sampling that do not require passing from samples of $q_{x_0}$ and $q_{x_1}$.

$\square$

## A.2 COMPOSING UNBALANCED FLOWS

**Proposition 2.** *Assuming $q_0$ and $q_1$ are densities with a bounded Lipschitz constant. A coupling $q(\mathbf{x}_0, \mathbf{x}_1) \propto q_0(\mathbf{x}_0)q_1(\mathbf{x}_1)\mathbb{I}_{\|\mathbf{x}_0-\mathbf{x}_1\|\leq\epsilon}$ with $\epsilon \to 0$ generates a marginal flow $\psi_1$ that has the form:*

$$\psi_1(\mathbf{x}_0) \approx \mathbf{x}_0 + C_{\epsilon,n}[q_0(\mathbf{x}_0)\nabla q_1(\mathbf{x}_0) - q_1(\mathbf{x}_0)\nabla q_0(\mathbf{x}_0)] \tag{12}$$

*where $C_{\epsilon,n}$ is a constant that depends on $\epsilon$ and the dimensionality of the space n.*

*Proof.* Let $\mathbf{u}_t(\mathbf{x}_t)$ be the vector field that generates the flow: $\frac{d}{dt}\psi_t(\mathbf{x}) = \mathbf{u}_t(\psi_t(\mathbf{x}))$. By definition $\mathbf{u}_t(\mathbf{x}_t)$ is the marginal vector field that optimizes the objective function of the defined Unbalanced FM. Therefore its derivative must be zero:

$$\mathbf{0} = \nabla_{\mathbf{u}_t(\mathbf{x}_t)}\mathbb{E}_{(\mathbf{x}_0,\mathbf{x}_1)\sim q}\left[\|\mathbf{u}_t(\mathbf{x}_t) - \mathbf{u}_t(\mathbf{x}_t|\mathbf{x}_1)\|^2\right]$$

$$= \nabla_{\mathbf{u}_t(\mathbf{x}_t)}\mathbb{E}_{(\mathbf{x}_0,\mathbf{x}_1)\sim q_0(\mathbf{x}_0)q_1(\mathbf{x}_1)\mathbb{I}_{\|\mathbf{x}_0-\mathbf{x}_1\|<\epsilon}/Z_q}\left[\|\mathbf{u}_t(\mathbf{x}_t) - (\mathbf{x}_1 - \mathbf{x}_0)\|^2\right]$$

Let $\delta = \mathbf{x}_1 - \mathbf{x}_0$. Expressing the expectation as a n-dimensional ball integral:

$$\int_{\|\delta\|\leq\epsilon} 2q_0(\mathbf{x}_t - \delta t)q_1(\mathbf{x}_t + \delta(1-t))(\mathbf{u}_t(\mathbf{x}_t) - \delta)\, Z_q\, d\delta = 0$$

Therefore:

$$\mathbf{u}_t(\mathbf{x}_t) = \int_{\|\delta\| \le \epsilon} \delta q_0(\mathbf{x}_t - \delta t) q_1(\mathbf{x}_t + \delta(1 - t)) \, d\delta$$

To approximate this integral as $\epsilon \to 0$, we expand $q_0$ and $q_1$ using a Taylor series expansion around $\mathbf{x}_t$:

$$q_0(\mathbf{x}_t - \delta t) \approx q_0(\mathbf{x}_t) - t\delta \cdot \nabla q_0(\mathbf{x}_t)$$

$$q_1(\mathbf{x}_t + \delta(1 - t)) \approx q_1(\mathbf{x}_t) + (1 - t)\delta \cdot \nabla q_1(\mathbf{x}_t)$$

Substituting the approximations into the integral:

$$\mathbf{u}_t(\mathbf{x}_t) \approx \int_{\|\delta\| \le \epsilon} \delta \left[ (q_0(\mathbf{x}_t) - t\delta \cdot \nabla q_0(\mathbf{x}_t)) (q_1(\mathbf{x}_t) + (1 - t)\delta \cdot \nabla q_1(\mathbf{x}_t)) \right] d\delta$$

Expanding the product inside the integral:

$$\mathbf{u}_t(\mathbf{x}_t) \approx \int_{\|\delta\| \le \epsilon} \delta \left[ q_0(\mathbf{x}_t) q_1(\mathbf{x}_t) + q_0(\mathbf{x}_t)(1 - t)\delta \cdot \nabla q_1(\mathbf{x}_t) - t\delta \cdot \nabla q_0(\mathbf{x}_t) q_1(\mathbf{x}_t) \right] d\delta$$

Higher-order terms involving $(\delta \cdot \nabla q_0)(\delta \cdot \nabla q_1)$ will be of order $\epsilon^2$ and can be neglected as $\epsilon \to 0$.

Since the integral is over a symmetric ball, terms with odd powers of $\delta$ vanish. Hence, only the cross terms contribute:

$$\mathbf{u}_t(\mathbf{x}_t) \approx \int_{\|\delta\| \le \epsilon} \delta \left[ (1 - t) q_0(\mathbf{x}_t) \delta \cdot \nabla q_1(\mathbf{x}_t) - t q_1(\mathbf{x}_t) \delta \cdot \nabla q_0(\mathbf{x}_t) \right] d\delta$$

Using the symmetry and volume element:

$$\int_{\|\delta\| \le \epsilon} \delta_i \delta_j \, d\delta = \frac{\epsilon^{n+2}}{n+2} \delta_{ij} \frac{\pi^{n/2}}{\Gamma(n/2 + 1)}$$

Therefore:

$$\mathbf{u}_t(\mathbf{x}_t) \approx \frac{\epsilon^{n+2}}{n+2} \frac{\pi^{n/2}}{\Gamma(n/2 + 1)} \left[ (1 - t) q_0(\mathbf{x}_t) \nabla q_1(\mathbf{x}_t) - t q_1(\mathbf{x}_t) \nabla q_0(\mathbf{x}_t) \right]$$

To determine the residual transformation brought by the full marginal flow, we need to integrate the vector field $\mathbf{u}_t(\mathbf{x}_t)$ over the path. Integrating over the path is challenging, but since we have restricted the flows to be smaller than $\epsilon$, the Lipschitz constant of $\mathbf{u}_t$ is bounded (because the Lipschitz constants of $q_0$ and $q_1$ are bounded), and we are taking $\epsilon \to 0$. The change in $\mathbf{u}_t$ within the flow will be negligible. Therefore, we can approximate the marginal flow as:

$$\psi_1(\mathbf{x}_0) \approx \mathbf{x}_0 + \int_{t=0}^{1} \frac{\pi^{n/2} \epsilon^{n+2}}{(n+2)\Gamma(n/2 + 1)} \left[ (1 - t) q_0(\mathbf{x}_0) \nabla q_1(\mathbf{x}_0) - t q_1(\mathbf{x}_0) \nabla q_0(\mathbf{x}_0) \right] dt$$

Evaluating the integral:

$$\psi_1(\mathbf{x}_0) \approx \mathbf{x}_0 + \frac{\pi^{n/2} \epsilon^{n+2}}{(n+2)\Gamma(n/2 + 1)} \left[ -\frac{t^2}{2} q_1(\mathbf{x}_0) \nabla q_0(\mathbf{x}_0) - \frac{(1 - t)^2}{2} q_0(\mathbf{x}_0) \nabla q_1(\mathbf{x}_0) \right]_0^1$$

$$\psi_1(\mathbf{x}_0) \approx \mathbf{x}_0 + \frac{\pi^{n/2} \epsilon^{n+2}}{2(n+2)\Gamma(n/2 + 1)} \left( q_0(\mathbf{x}_0) \nabla q_1(\mathbf{x}_0) - q_1(\mathbf{x}_0) \nabla q_0(\mathbf{x}_0) \right)$$

$\square$

# B TRAINING AND INFERENCE

In this section, we present the training and inference procedures for our manifold docking (Algorithms 3 and4) and relaxation models (Algorithms 5 and6). We refer to unbound protein structures as apo structures, and the bound structures as holo structures. Recall that our goal is to learn a distribution over holo structures, given the apo structure and a seed conformation of the ligand. Similar to (Corso et al., 2022), we are in a setting, where traditional generative modeling where one has access to multiple samples from the same data distribution, we only have a single $(\mathbf{x}^*, \mathbf{y}_{\text{apo}}, \mathbf{y}_{\text{holo}})$ per protein-ligand complex. This implies that the training loop (Algorithms 3 and5) now proceeds over different distributions, along with a single sample from that distribution. This sample is then accepted or rejected depending on the cutoff $c_{\text{dock}}$, thus inducing an unbalanced coupling and flow.

---

**Algorithm 3:** TRAINING EPOCH: MANIFOLD DOCKING

---

**Input:** Training Pairs $\{(\mathbf{x}^*, \mathbf{y}_{\text{apo}}, \mathbf{y}_{\text{holo}})\}$; RDKit predictions $\{\mathbf{c}\}$, RMSD cutoff $c_{\text{dock}}$
**Input:** Pocket radius $r$; Pocket Buffer $b$
**Input:** C$\alpha$ operator $[\cdot]_{C\alpha}$

**foreach** $\mathbf{c}, \mathbf{x}^*, \mathbf{y}_{apo}, \mathbf{y}_{holo}$ **do**

    Let $\mathbf{x}_0 \leftarrow \arg\min_{\mathbf{x}} \text{RMSD}(\mathbf{x}^*, \mathbf{x})$
    $\mathbf{y}_{\text{center}}, \{i\}_{\text{pocket}} = \text{EXTRACTPOCKET}(\mathbf{y}_{\text{apo}}, \mathbf{y}_{\text{holo}}, r, b)$
    $\mathbf{y}_{\text{apo}} \leftarrow \text{RMSDALIGN}(\mathbf{y}_{\text{apo}}, \mathbf{y}_{\text{holo}}, \{i\}_{\text{pocket}})$
    **if** $\text{RMSD}(\mathbf{y}_{apo}, \mathbf{y}_{holo}) > c_{dock}$ **then**
      |   **continue**
    **else**

        $\mathbf{y}_{\text{apo}}^{\text{FA}}, \Delta R^{\text{bb}} \leftarrow \text{FRAMEALIGN}(\mathbf{y}_{\text{apo}}, \mathbf{y}_{\text{holo}})$
        $\mathbf{y}_{\text{apo}}^{\text{FA,SC}}, \Delta \theta^{\text{sc}} \leftarrow \text{SCCONFMATCH}(\mathbf{y}_{\text{apo}}^{\text{FA}}, \mathbf{y}_{\text{holo}})$
        Sample $t \sim \mathcal{U}(0, 1)$

        // Ligand Diffusion
        Sample $\Delta r, \Delta R, \Delta \theta$ from diffusion kernels $p_t^{\text{tr}}(\cdot|0), p_t^{\text{rot}}(\cdot|0), p_t^{\text{tor}}(\cdot|0)$
        Compute $\mathbf{x}_t$ by applying $(\Delta r, \Delta R, \Delta \theta)$ to $\mathbf{x}_0$

        // Protein Flow
        $t^{\text{sc}}, t_{\text{rot}}^{\text{bb}}, t_{\text{tr}}^{\text{bb}} = \text{COMPUTETIME}(t, \alpha^{\text{sc}}, \alpha_{\text{rot}}^{\text{bb}}, \alpha_{\text{tr}}^{\text{bb}})$
        Interpolate $\Delta r_t^{\text{bb}} \leftarrow [\mathbf{y}^{apo}]_{C\alpha} \cdot (1 - t) + [\mathbf{y}^{holo}]_{C\alpha} \cdot t$
        $u_t^{\text{tr,bb}}(\cdot|z) \leftarrow [\mathbf{y}^{holo}]_{C\alpha} - [\mathbf{y}^{apo}]_{C\alpha}$

        Interpolate $\Delta R_t^{\text{bb}} \leftarrow \exp\left(t_{\text{rot}}^{\text{bb}} \log(\Delta R^{\text{bb}})\right)$
        $u_t^{\text{rot,bb}}(\cdot|z) \leftarrow \dfrac{\log_{\Delta R_t^{\text{bb}}}(\Delta R^{\text{bb}})}{1 - t_{\text{rot}}^{\text{bb}}}$

        Interpolate $\Delta \theta_t^{\text{sc}} \leftarrow \exp\left(t^{\text{sc}} \log(\Delta \theta^{\text{sc}})\right)$
        $u_t^{\text{sc}}(\cdot|z) \leftarrow \dfrac{\log_{\Delta \theta_t^{\text{sc}}}(\Delta \theta^{\text{sc}})}{1 - t^{\text{sc}}}$
        Compute $\mathbf{y}_t$ by applying $\left(\Delta r^{\text{bb}}, \Delta R^{\text{bb}}, \Delta \theta^{\text{sc}}\right)$ to $\mathbf{y}_{\text{apo}}$

        Predict scores and drifts $\alpha, \beta, \gamma, \delta, \epsilon, \eta \leftarrow s(\mathbf{x}_t, \mathbf{y}_t, t)$
        // Ligand Loss
        $\mathcal{L}_{\text{lig}} = \|\alpha - \nabla \log p_t^{\text{tr}}(\cdot|0)\|^2 + \|\beta - \nabla \log p_t^{\text{rot}}(\cdot|0)\|^2 + \|\gamma - \nabla \log p_t^{\text{tor}}(\cdot|0)\|^2$
        // Protein Loss
        $\mathcal{L}_{\text{prot}} = \|\delta - u_t^{\text{tr,bb}}(\cdot|z)\|^2 + \|\epsilon - u_t^{\text{rot,bb}}(\cdot|z)\|^2 + \|\eta - u_t^{\text{sc}}(\cdot|z)\|^2$
        Apply optimization step on $\mathcal{L} = \mathcal{L}_{\text{prot}} + \mathcal{L}_{\text{lig}}$
    **end**

**end**

---

**Pocket Extraction.** As our focus is on the flexible protein docking task, we first extract the protein pocket given apo and holo-structures. We define the pocket residues as all residues in the holo-

---

**Algorithm 4:** INFERENCE: MANIFOLD DOCKING

---

**Input:** RDKit predictions $\{\mathbf{c}\}$, Apo structure $\mathbf{y}_{\text{apo}}$ of the protein pocket
**Input:** Inference Steps $N$
Sample $\Theta_N \sim \mathcal{U}(SO(2)^m)$, $R_N \sim \mathcal{U}(SO(3))$, $r_n \sim \mathcal{N}(0, \sigma_{\text{tr}}^2)$
Apply $\Theta_N, R_N, r_n$ to $c$ to get $\mathbf{x}_N$
Set $\mathbf{y}_N \leftarrow \mathbf{y}_{\text{apo}}$
$\Delta t \leftarrow 1/N$
**for** $n \leftarrow N$ **to** $1$ **do**
    $t \leftarrow n/N$
    Predict scores and drifts $\alpha, \beta, \gamma, \delta, \epsilon, \eta \leftarrow s(\mathbf{x}_n, \mathbf{y}_n, t)$

    // Ligand Updates
    $\Delta\sigma_{\text{tr}}^2 = \sigma_{\text{tr}}^2(n/N) - \sigma_{\text{tr}}^2((n-1)/N)$
    $\Delta\sigma_{\text{rot}}^2 = \sigma_{\text{rot}}^2(n/N) - \sigma_{\text{rot}}^2((n-1)/N)$
    $\Delta\sigma_{\text{tor}}^2 = \sigma_{\text{tor}}^2(n/N) - \sigma_{\text{tor}}^2((n-1)/N)$
    Sample $\mathbf{z}_{\text{tr}}, \mathbf{z}_{\text{rot}}, \mathbf{z}_{\text{tor}}$ from $\mathcal{N}(0, \sigma_{\text{tr}}^2), \mathcal{N}(0, \sigma_{\text{rot}}^2), \mathcal{N}(0, \sigma_{\text{tor}}^2)$
    Apply $(\alpha, \beta, \gamma)$ to $\mathbf{x}_n$ to get $\mathbf{x}_{n-1}$

    // Protein Updates
    $\Delta r_n^{\text{bb}} \leftarrow \delta \cdot \Delta t$
    $\Delta R_n^{\text{bb}} \leftarrow \epsilon \cdot \Delta t$
    $\Delta\theta_n^{\text{sc}} \leftarrow \eta \cdot \Delta t$
    Apply $\left(\Delta r_n^{\text{bb}}, \Delta R_t^{\text{bb}}, \Delta\theta_n^{\text{sc}}\right)$ to $\mathbf{y}_n$ to get $\mathbf{y}_{n-1}$
**end**

---

structure that have at least one heavy atom within 5Å of any ligand atom. Given these pocket residues, the pocket center is defined based on the positions of the $C\alpha$ atom in the apo structure. To construct the geometric graphs (Appendix D), we also use the residues which have a $C\alpha$ atom within 20Å of the pocket center. This additional buffer is added to improve the model's robustness to exact pocket definitions, and also add geometric information from the pocket neighborhood.

**Aligning Apo-Holo Frames.** A residue frame (Jumper et al., 2021), is characterized by a tuple $(R, t) \in SE(3)$, where the rotation $R$ is about the origin of the residue, and $t$ specifies the position of the $C\alpha$ atom. Before applying the conformer matching step (explained below) to the protein sidechains, we align the frames of the apo and holo structures, by computing the rotation that aligns the $N - C\alpha$ vectors of the corresponding residues. The alignment will not be perfect owing to differences in the bond lengths and bond angles between the computationally generated and ground truth structures, but provides the closest modification of the apo structure backbone to the holo-structure one.

**Conformer Matching.** For both the ligand and the protein sidechains, we apply the conformer matching procedures in (Jing et al., 2022) and (Plainer et al., 2023), where, given the local structures from computational methods, we find the closest (in an RMSD sense) structure to the ground truth by modifying the appropriate torsion angles. The conformer matching procedure is employed to prevent a distribution shift between training and inference in the local structures that are considered rigid in the manifold docking process. To elaborate, the local structures (such as bond lengths and bond angles) vary between RDKit (for ligands) and ESMFold (for proteins) generated structures, and their ground truth counterparts. If we train our models with ground truth local structures, this would cause a distribution shift at inference time, when we only have access to local structures, provided by RDKit and ESMFold.

### B.1 ENERGY LOSS FORMULATION

To construct the flat-bottom potential defined in Equation 10, for any pair of atoms (i, j), we define the lower and upper distances bounds $L_{i,j}$ and $U_{i,j}$ in the following manner:

    1. If the atom pair (i, j) represents a covalent bond between ligand atoms

---

**Algorithm 5:** TRAINING EPOCH: RELAXATION

---

**Input:** Training Pairs $\{(\mathbf{x}_{\text{dock}}, \mathbf{x}_{\text{holo}}, \mathbf{y}_{\text{dock}}, \mathbf{y}_{\text{holo}}, \{i\}_{\text{pocket}})\}$, RMSD cutoff $c_{\text{relax}}$

**foreach** $\mathbf{x}_{dock}, \mathbf{y}_{holo}, \mathbf{y}_{dock}, \mathbf{y}_{holo}, \{i\}_{pocket}$ **do**
    **if** $0.5 \cdot \text{RMSD}(\mathbf{y}_{dock}, \mathbf{y}_{holo}) + 0.5 \cdot \text{RMSD}(\mathbf{x}_{dock}, \mathbf{x}_{holo}) > c_{relax}$ **then**
        | **continue**
    **else**
        Sample $t \sim \mathcal{U}(0,1)$

        // Ligand Flow
        Interpolate $\Delta r_t^{\text{lig}} \leftarrow \mathbf{x}_{\text{dock}} \cdot (1-t) + \mathbf{x}_{\text{holo}} \cdot t$
        $u_t^{\text{lig}}(\cdot|z) \leftarrow \mathbf{x}_{\text{holo}} - \mathbf{x}_{\text{dock}}$
        Apply $\Delta r_t^{\text{lig}}$ to $\mathbf{x}_{\text{dock}}$ to get $\mathbf{x}_t$

        // Protein Flow
        Interpolate $\Delta r_t^{\text{prot}} \leftarrow \mathbf{y}_{\text{dock}} \cdot (1-t) + \mathbf{y}_{\text{holo}} \cdot t$
        $u_t^{\text{prot}}(\cdot|z) \leftarrow \mathbf{y}_{\text{holo}} - \mathbf{y}_{\text{dock}}$
        Apply $\Delta r_t^{\text{prot}}$ to $\mathbf{y}_{\text{dock}}$ to get $\mathbf{y}_t$

        // Make Prediction
        Predict drifts $\alpha, \beta \leftarrow s_r(\mathbf{x}_t, \mathbf{y}_t, t)$
        Apply $\alpha$ to $\mathbf{x}_{\text{dock}}$ to get $\hat{\mathbf{x}}_1$
        Apply $\beta$ to $\mathbf{y}_{\text{dock}}$ to get $\hat{\mathbf{y}}_1$
        $\hat{\mathbf{x}}_1, \hat{\mathbf{y}}_1 \leftarrow \text{RMSDALIGN}(\hat{\mathbf{x}}_1, \hat{\mathbf{y}}_1, \mathbf{y}_{\text{holo}}, \{i\}_{\text{pocket}})$

        // Compute Loss
        $\mathcal{L}_{\text{lig}} = \frac{1}{N} \sum_i^N \|\hat{\mathbf{x}}_1^{(i)} - \mathbf{x}_{\text{holo}}^{(i)}\|^2$
        $\mathcal{L}_{\text{prot}} = \frac{1}{M} \sum_i^M \|\hat{\mathbf{y}}_1^{(i)} - \mathbf{y}_{\text{holo}}^{(i)}\|^2$
        $\mathcal{L}_{\text{internal\_energy}} = \sum_{i,j} \max\left(\left\|\hat{\mathbf{x}}_1^{(i)} - \hat{\mathbf{x}}_1^{(j)}\right\| - U_{i,j}, 0\right) + \max\left(L_{i,j} - \left\|\hat{\mathbf{x}}_1^{(i)} - \hat{\mathbf{x}}_1^{(j)}\right\|, 0\right)$
        $\mathcal{L}_{\text{cross\_energy}} = \sum_{i,j} \max\left(L_{i,j} - \left\|\hat{\mathbf{x}}_1^{(i)} - \hat{\mathbf{y}}_1^{(j)}\right\|, 0\right)$
        $\mathcal{L} = \lambda^{\text{lig}} \mathcal{L}_{\text{lig}} + \lambda^{\text{atom}} \mathcal{L}_{\text{atom}} + \lambda_t^{\text{energy}}(\mathcal{L}_{\text{internal\_energy}} + \mathcal{L}_{\text{cross\_energy}})$
    **end**
**end**

---

- $L_{i,j}$ is set to 0.75 times the lower distance bound used by RDKit
- $U_{i,j}$ is set to 1.25 times the upper distance bound used by RDKit

2. If the atom pair (i, j) represents two substituents of the same central ligand atom

- $L_{i,j}$ is set to 0.75 times the lower distance bound used by RDKit
- $U_{i,j}$ is set to 1.25 times the upper distance bound used by RDKit

3. If the atom pair (i, j) represents two ligand atoms that are neither covalently bonded nor bonded to the same central atom

- $L_{i,j}$ is set to 0.8 times the lower distance bound used by RDKit
- No upper bound is used

4. If the atom pair (i, j) includes one protein and one ligand atom

- $L_{i,j}$ is set to the sum of each atom's Van der Waal's radius used by RDKit minus 0.75
- No upper bound is used

5. If the atom pair (i, j) represents two protein atoms

- No lower bound is used
- No upper bound is used

Under this formulation, any conformation for which the potential is zero will pass the following PoseBusters validity checks as defined in Buttenschoen et al. (2024):

---

**Algorithm 6:** INFERENCE: RELAXATION

---

**Input:** Manifold Docking Predictions $(\mathbf{x}_{\text{dock}}, \mathbf{y}_{\text{dock}})$
**Input:** Inference Steps $N$
Set $\mathbf{x}_1 \leftarrow \mathbf{x}_{\text{dock}}$
Set $\mathbf{y}_1 \leftarrow \mathbf{y}_{\text{dock}}$
$\Delta t \leftarrow 1/N$
**for** $n \leftarrow 0 \, \mathbf{to} \, N - 1$ **do**
$\quad$ $t \leftarrow n/N$
$\quad$ Predict scores $\alpha, \beta \leftarrow s(\mathbf{x}_n, \mathbf{y}_n, t)$
$\quad$ Apply $\alpha$ to $\mathbf{x}_n$ to get $\mathbf{x}_{n_1}$
$\quad$ Apply $\beta$ to $\mathbf{y}_n$ to get $\mathbf{y}_{n_1}$
**end**

---

- **Bond length validity** due to condition (1)
- **Bond angle validity** due to condition (2)
- **Internal steric clash** due to condition (3)
- **Minimum protein-ligand distance** due to condition (4)

During training, we scale the energy loss term from a weight of 0 at $t = 0$ to 1 at $t = 1$ with the exponential schedule: $\lambda_t^{\text{energy}} = \frac{e^{\alpha t} - 1}{e^{\alpha} - 1}$ and $\alpha = -2.5$. Additionally, since the RMSD of the ligand is typically greater than that of the protein in docked poses, we further scale the ligand and protein losses with weights $\lambda^{\text{lig}} = 1.0$ and $\lambda^{\text{prot}}$ respectively.

## C  MODEL ARCHITECTURE

We use message passing networks based on tensor products of irreducible representations (irreps) of $SO(3)$, implemented with the `e3nn` library.

**Graph Construction.**  We represent structures as geometric heterogeneous graphs, with nodes comprising ligand heavy atoms, receptor residues in the pocket and neighborhood (located at the position of C$\alpha$ atoms), and the heavy atoms of the pocket residues. We chose to only model the heavy atoms of the pocket residues for two reasons - i) this provides a useful sparsity constraint for computational and memory efficiency, and ii) typically, most of the conformational changes in the protein involve the pocket atoms, and modeling this explicitly would facilitate downstream applications such as affinity prediction. We also adopt different cutoffs depending on the types of nodes being connected, largely following (Corso et al., 2022):

1. Ligand atoms-ligand atoms, receptor atoms-receptor atoms, and ligand atom-receptor atom interactions use a cutoff of 5Å. Covalent bonds between ligand atoms are explicitly modeled with initial edge embeddings to reflect the type of bond. For receptor atoms, we limit the maximum number of neighbors to 12.

2. For receptor residue interactions, we use a distance cutoff of 15Å, with a maximum neighbor limit of 24.

3. For interactions between ligand atoms and receptor residues, unlike (Corso et al., 2022), we found using the dynamic cutoff based on the ligand translation noise to cause NaNs during training, possibly due to missing connections. We thus used a distance cutoff of 80Å between ligand atoms and receptor residues.

4. Receptor pocket atoms are also connected to their corresponding residues.

**Featurization.**  We adopted the same featurization as DIFFDOCK, using the residue type and the embeddings with ESM2 Language model for the residues, the atom type, and other chemical properties for the ligand and receptor atoms. The relaxation model additionally uses the Van der Waals radii as features for ligand and receptor atoms as well as the RDKit upper and lower distance geometry bounds as features for ligand edges.

**Manifold Docking and Relaxation** We retain the core architecture of DIFFDOCK (Corso et al., 2022), with the tensor product convolution-based message-passing layers, followed by a convolution with the center of mass to predict the rotational and translation scores for the ligand. For the torsion angles in the ligand and sidechain torsion angles in the protein, we use the pseudotorque layer from (Jing et al., 2022), adapted accordingly for the sidechains. To predict the rotation and translation flows for the residues (which are $SE(3)$ equivariant), we use a linear layer that transforms the irreps of the residue embeddings to a single odd and even vector (one for each flow). Since the residues constitute a coarse-grained representation of the protein, we sum the odd and even vector representations to obtain the predictions. For the relaxation model, we predict a single vector for each ligand and protein atom. The magnitudes of the predictions are then adjusted with an MLP.

**Confidence Model.** The embedding layers for the confidence model follow the same architecture as the manifold docking and relaxation. The aggregated ligand, receptor residue, and receptor atom embeddings are concatenated, and updated with an MLP to predict the final confidence (a $SE(3)$ invariant output).

## D EXPERIMENTAL DETAILS

The code used for running our experiments can be found at `https://github.com/vsomnath/flexdock`.

**Data.** For training our models, we use the PDBBind dataset (Liu et al., 2017) whose complexes were extracted from the PDB. Following (Stärk et al., 2022; Corso et al., 2022), we adopt the time-based split of PDBBind, where the 17k complexes before 2019 were divided into training and validation sets, while the 363 complexes after 2019 form the test set. We download the PDBBind data as it is provided by EquiBind Stärk et al. (2022). These files are first processed by `PDBFixer` from the `OpenMM` toolbox (Eastman et al., 2017), to replace non standard residues and add missing atoms. We then used the `PDBFixer` processed files to extract the protein sequence, and predict its structure with ESMFold (Lin et al., 2022). The ESMFold generated files are also processed by `PDBFixer` to add missing atoms such as terminal oxygens, at the end of a chain. These processed files now constitute our apo structures, while the processed analogues from PDBBind constitute our holo structures. We further remove hydrogen atoms while aligning the apo and holo structures.

To measure the validity and physicality of generated poses, we also use the PoseBusters suite of tests (Buttenschoen et al., 2024), which evaluate i) chemical validity, ii) intramolecular validity for the ligand poses based on energetics and geometry, and iii) intermolecular interactions and steric clashes between protein and ligand.

**Metrics.** To evaluate the generated ligand and protein pocket poses, we compute the RMSD between the predicted and ground truth poses after alignment. This alignment is computed based on the Kabsch alignment between the atoms in the protein pocket, in the ground truth and predicted poses. To account for permutation symmetries in the ligand, we use the symmetry-corrected RMSD of sPyRMSD. For the ligand, besides the median RMSD, we report the % of RMSDs below 2Å, which is a commonly adopted metric for judging the quality of docking predictions (Alhossary et al., 2015; Hassan et al., 2017; McNutt et al., 2021). For the protein pocket atoms, besides the median RMSD, we report the % of RMSDs below 1Å, where we chose the 1Å cutoff, typically treated as atomic accuracy.

**Training Details.** For our manifold docking model (75.3 M parameters), we use an exponential moving average of weights (EMA) during training, which is updated at every optimization step, with a decay factor of 0.999. We train the model on 4 RTX A6000 GPUs, with a batch size of 4 per GPU. Every 10 epochs, we run inference for 20 steps with the EMA weights on 500 complexes in the validation set, and save the model with the largest percentage of ligand RMSDs < 2Å. The initial learning rate of the model is 0.001, which is updated with a learning rate scheduler with decay 0.7 if the percentage of complexes with ligand RMSDs < 2Å does not improve over 30 epochs. We train our model for 600 epochs, after which we did not observe a noticeable increase in ligand RMSDs < 2Å metric. We use the ADAM optimizer for all our models.

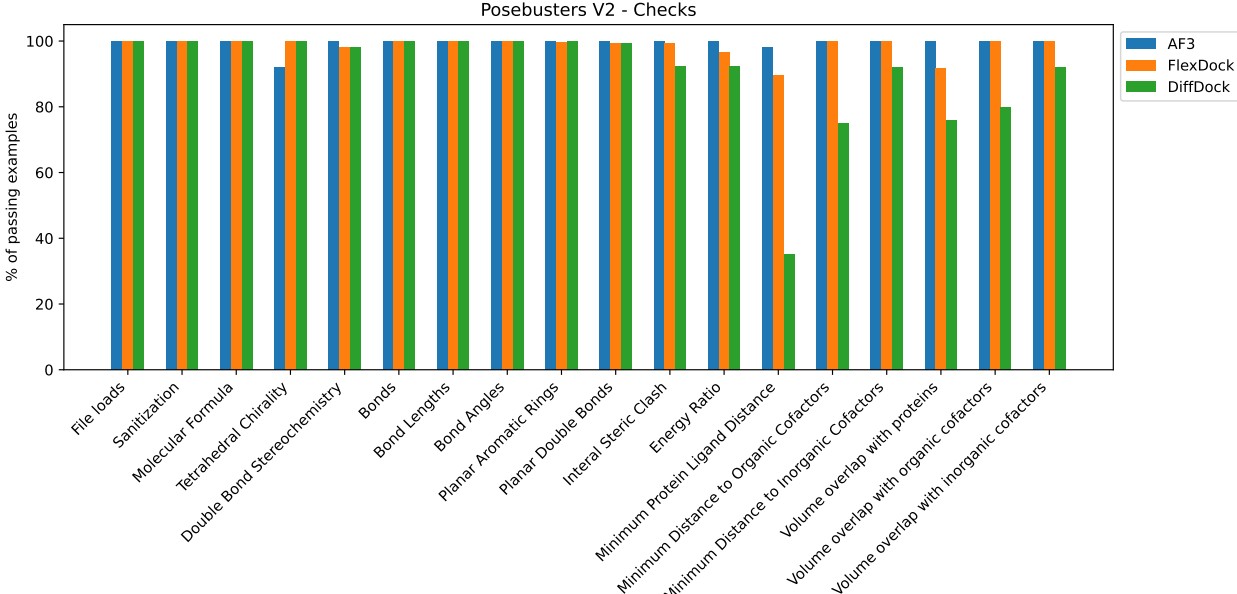

Figure 6: Detailed comparison of the validity checks offered in PoseBusters V2 benchmark

For the confidence model, we use a smaller version of the manifold docking model 4 M parameters to generate 20 poses (both ligand and protein) per training complex. For the ligand, we assign label 1 if the RMSDS between predicted (after alignment) and ground truth pose is <2Å, while for protein pocket atoms, we adopt an RMSD cutoff of 1Å. We train the confidence model for around 100 epochs, and save the model with the best accuracy. We found the model predicting only the ligand pose confidence to offer the best tradeoff between ligand and pocket atom prediction confidence.

**Runtimes.** Similar to other ML docking baselines, we measure runtimes for the manifold docking and confidence model. These runtimes are calculated on a single RTX A100 80GB GPU, with the preprocessing steps entailing ESM2 embedding generation and RDKit conformer generation. The geometric graphs are generated on the fly as part of the model and thus are already included in the runtimes.

# E  ADDITIONAL ANALYSIS

## E.1  COMPARISONS ON POSEBUSTERS BENCHMARK

We include a stratification of different validity checks adopted in PoseBusters, comparing against ALPHAFOLD3 and DIFFDOCK in Figure 6 . The numbers for ALPHAFOLD3 and DIFFDOCK were taken from (Abramson et al., 2024). We find that FLEXDOCK generates highly valid molecules with very low computational overhead ( 0.3s), and has a significant improvement from DIFFDOCK on the intermolecular validity checks.

## E.2  COMPARING RMSD DISTRIBUTIONS

**Training and Test Distributions** In Figure 7, we plot the all-atom (AA) RMSDs between ESM-FOLD and holo structures in the training and test sets. While the AA-RMSDs is largely within 4A, there is a very long tail of significantly larger distances which impact the expected squared coupling distance when using Flow Matching.

**Large Conformational Changes.** In Figure 8, we plot the AA-RMSDs between the ESMFold and holo structures, and also between the corresponding predicted and holo structures for large conformational changes (> 4Å). We find that FLEXDOCK is able to shift the distributions towards

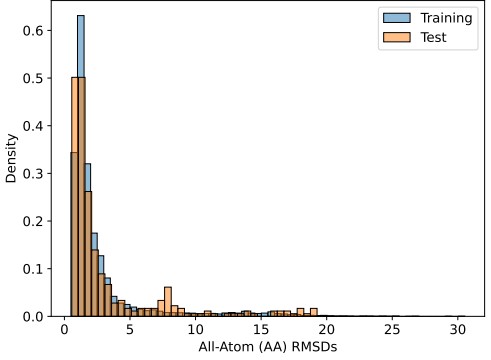 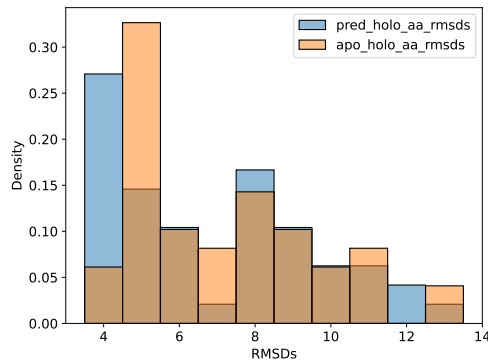

Figure 7: Distributions of AA-RMSDs between ESMFold and holo structures in the training and test sets.

Figure 8: RMSD Distributions for Large Conformational Changes (>4Å)

lower RMSDs compared to the initial distribution. This is reasonable since the model updates the apo structure to the closest structure based on the learned underlying transport. Additionally, we find that 77.5% of poses pass the Posebusters validity checks, indicating the ability of our model to find physically meaningful poses, even in this extreme setting.

**ESMFlow vs ESMFold**   In Figure 8, we plot the AA-RMSDs between the apo and holo structures, where the apo structures are generated either using ESMFold or its flow matching counterpart ESMFlow (Jing et al., 2024). The distributions look largely similar, with most of the RMSDs within 5A, followed by a long tail of larger RMSDs. This also highlights the limited ability of current methods to capture the diversity in the conformational space of apo protein structures.

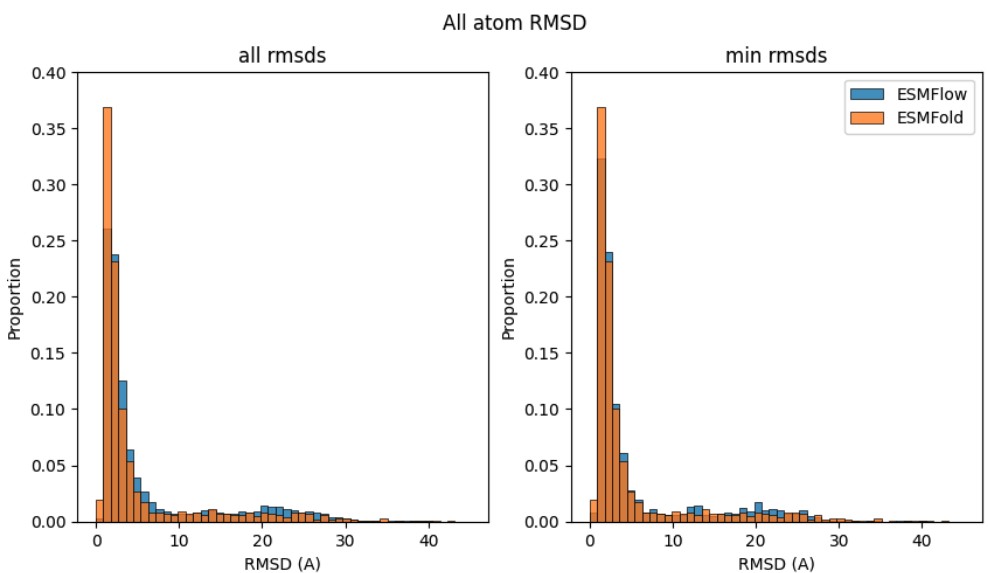

Figure 9: AA-RMSD distributions between apo and holo structures, with apo structures generated using ESMFold or ESMFlow

### E.3   SAMPLE EFFICIENCY

In practice, rather than the actual sampling efficiency, one is often interested in indirect measurements of the same, and using that to compare different methods. These indirect measurements are based

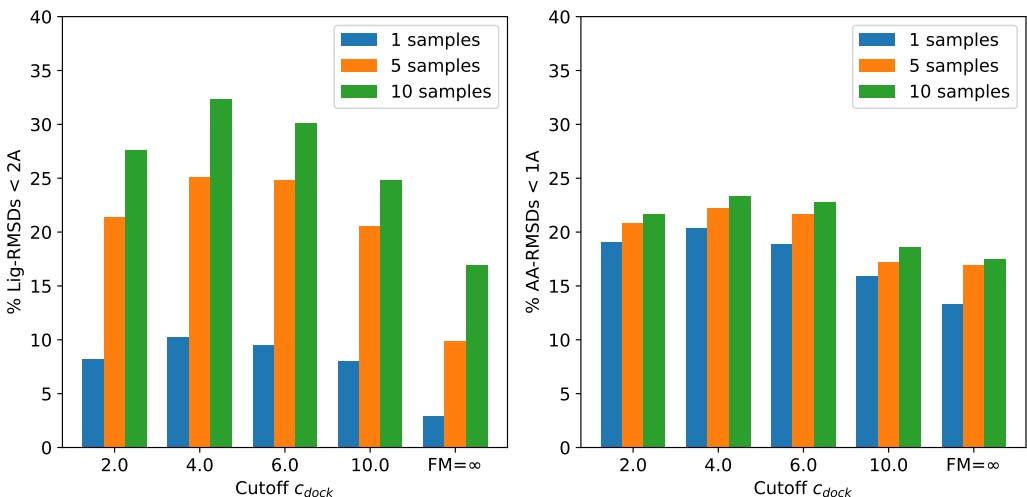

Figure 10: Sample Efficiency for different cutoffs $c_{\mathrm{dock}}$

on performance achieves on desired metrics, with a fixed number of samples. In Figure 10, we plot the best achievable performance for each $c_{\mathrm{dock}}$ for $1, 5, 10$ samples. Comparing FM with UFM (4A), where $c_{\mathrm{dock}} = 4.0$ for instance, we can clearly see that even with 10 samples, FM achieves a worse performance on ligand docking, compared to UFM (4A), with $5$ samples, making UFM (4A) more sample efficient.

# F  VISUALIZED EXAMPLES

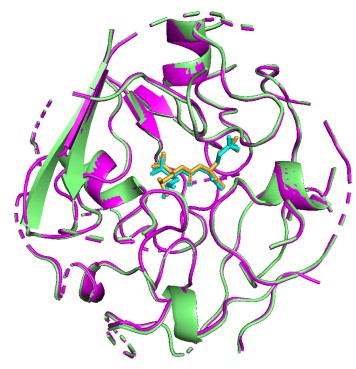

(a) 6FE5, L-RMSD: 0.66, AA-RMSD: 0.72, APO-HOLO-RMSD: 3.93

(b) 6OD6, L-RMSD: 0.61, AA-RMSD: 0.81, APO-HOLO-RMSD: 1.67

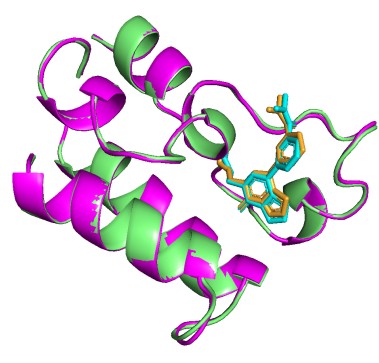

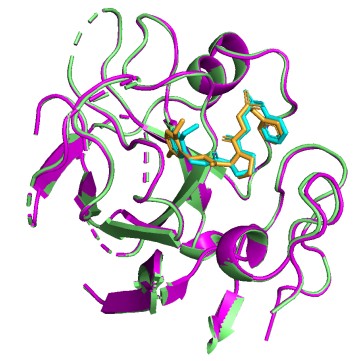

(c) 6BQD, L-RMSD: 0.51, AA-RMSD: 0.42, APO-HOLO-RMSD: 0.56

(d) 6ROT, L-RMSD: 0.71, AA-RMSD: 0.57, APO-HOLO-RMSD: 0.94

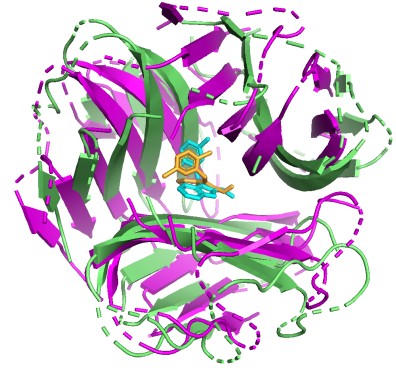

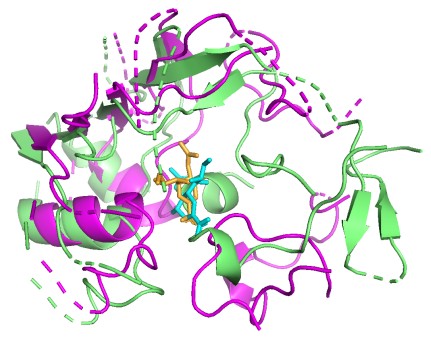

(e) 6OOY, L-RMSD: 3.97, AA-RMSD: 8.75

(f) 6CKL, L-RMSD: 3.51, AA-RMSD: 9.01

Figure 11: Visualized predictions for the complexes 6FE5, 6OD6, 6BQD, 6ROT, 6OOY, and 6CKL in the PDBBind test dataset. L-RMSD and AA-RMSD denote Ligand RMSD and All-Atom RMSD respectively, after aligning the predicted and ground truth pockets. Color Code: Holo protein (in lime), predicted protein (magenta), true ligand (cyan) and predicted ligand (orange).

