# OpenReview forum: "Composing Unbalanced Flows for Flexible Docking and Relaxation"
_ICLR.cc/2025/Conference — ICLR 2025 Oral_

### Official Review · Reviewer_ZfZb · 2024-11-02

**Soundness:** 3
**Presentation:** 3
**Contribution:** 3
**Rating:** 10
**Confidence:** 5

**Summary:**

This paper proposes Unbalance Flow Matching and demonstrates its application to flexible docking, and enhances its performance from 30% to 73%. For those who have attempted the traditional flow on flexible docking or protein domain structural changes, they must have encoutered the same problem addresses in this paper. It offers a good solution backed by theoretical insights and supporting proofs.

**Strengths:**

- It proposes the Unbalance Flow Matching (UFM), which extends the traditional flow matching with the coupling q by performing some filtering (e.g., RMSD) in practice.
- The proposed two-stage flow matching process, consisting of a rough docking stage called “manifold docking” and a more precise “relaxation” stage, effectively boosts the PB valid benchmark to 73% while maintaining reasonable computational costs.

**Weaknesses:**

- I do not see major weakness in this paper, but have several questions as follows.

**Questions:**

- In Section 3, the authors mention challenges in applying flow matching directly to flexible docking:

    > In fact, although the different conformational states of the protein do not change significantly upon ligand binding, their relative weights are often notably
    altered.
    >

    What is the **relative weights** meaning?

- The proposed second stage is to perform the relaxation by deep learning model. Maybe it is to give an end-to-end solution for more precise docking, as shown in Table 2. But I strongly recommend to conduct the minimisation (relaxation) using force fields instead as what is done in PoseBusters (Section 2.5 and 3.3 in PoseBusters paper). The comparison between PB valid and runtime consumption is preferred.
- For the apo structure, as mentioned in 4.1:
>given the relative inaccuracy of computational models to sample multiple protein conformations.
>
I’m wondering why the ensembles generated by, for example, AlphaFlow, is inaccuracy. Since it is sampled from the MD trajectories and, in principle, it should provide more accurate structures than the addition of small Gaussian noise on apo structure.
- The unbalanced pattern is pinned across the training. However, is there a potential risk that, for example, there are some apo structures all have RMSD exceeding 4Å from the holo structures? Under such situation, they are all been filtered out and the sampling would fail. Should the traditional flow and unbalanced flow been given a ratio during the training process (like 5/5)?
- The comparison between AF3 on PB is preferred (extended Data Fig.4).

---

> ### Author Response · Authors · 2024-11-22
> **Response to reviewer ZfZb 1/2**
>
> Thank you for the time dedicated to reviewing our work and the constructive feedback. Below we respond to each one of your questions or comments and integrate the new experiments and clarifications in the updated version of the manuscript.
>
> *What is the relative weights meaning?*
>
> Relative weights here refer to the relative probabilities of the two conformational states. At a high level, the distributions over apo and holo structures tend to have the same support, but the latter assigns a higher probability to structures that can accommodate the ligand.
>
> *The proposed second stage is to perform the relaxation by deep learning model. Maybe it is to give an end-to-end solution for more precise docking, as shown in Table 2. But I strongly recommend to conduct the minimisation (relaxation) using force fields instead as what is done in PoseBusters (Section 2.5 and 3.3 in PoseBusters paper). The comparison between PB valid and runtime consumption is preferred.*
>
> Our motivation for using a second model for structure relaxation stems from i)  the theory of composing unbalanced flows (Proposition 2), and ii) the slow relaxation times that current methods offer. Following your suggestions, we also conducted a relaxation using Posebusters energy minimization tool (which relies on the OpenMM package and the addition of a number of constraints), with the results presented in the plots below:
>
> | Method                  | Time (System Prep + Relaxation) | PB-Valid (%) |
> |-------------------------|----------------------------------|--------------|
> | PoseBusters Minimization | 155s                           | 68.4%        |
> | Ours                    | 0.3s                           | 72.9%        |
>
> This highlights the efficiency and effectiveness of our relaxation framework. We added this discussion and comparison to Section 5.
>
> *For the apo structure, as mentioned in 4.1: given the relative inaccuracy of computational models to sample multiple protein conformations. I’m wondering why the ensembles generated by, for example, AlphaFlow, is inaccuracy. Since it is sampled from the MD trajectories and, in principle, it should provide more accurate structures than the addition of small Gaussian noise on apo structure.*
>
> AlphaFlow (and ESMFlow) are both trained on small (300ns) MD trajectories, hence its ability to sample the conformational space is limited. Our comment on inaccuracy was more reflective of AlphaFlow’s (in)-ability to sample the conformational space, and we have clarified the statement to reflect this.
>
> On small training runs, we didn’t notice a significant difference in performance between using ESMFlow-MD structures vs a single-ESMFold. We report some findings around this in Appendix E showing that even taking multiple samples from ESMFlow does not significantly improve the minimum protein RMSD.
>
> *The unbalanced pattern is pinned across the training. However, is there a potential risk that, for example, there are some apo structures all have RMSD exceeding 4Å from the holo structures? Under such situation, they are all been filtered out and the sampling would fail. Should the traditional flow and unbalanced flow been given a ratio during the training process (like 5/5)?*
>
> This situation is possible but somewhat rare because, while the distribution changes upon binding, the poses that the protein takes when bound to the ligand are relatively close to poses that have good energetics even when unbound. However, when this happens during training, with this coupling distribution, this example will not be used. During inference the model will try to find a good protein conformation in the vicinity of the apo structure for the ligand to bind.
>
> In fact, we observe that when only looking at the structures where the distance between apo and holo is greater than 4A the distribution of generated poses by the model does tend to shift towards lower RMSD, but the shift is limited. However, the PoseBuster checks are still good on these poses (77.5% passing) suggesting that the poses are still physical. We added a full analysis on this in Appendix E.
>
> In this work, we only focused on simple couplings which would accept or reject samples in a binary manner. That said, depending on the task and dataset, one could instead use couplings that accept samples with large RMSDs with a low probability, depending on the specified cutoff. This would also tie into the reviewer's suggestion of ratios between unbalanced and traditional flow matching.

---

> ### Author Response · Authors · 2024-11-22
> **Response to reviewer ZfZb 2/2**
>
> *The comparison between AF3 on PB is preferred (extended Data Fig.4).*
>
> Thank you for the suggestion! We have now added the plots for Posebusters V2 in Section 5 and the validity checks in Appendix E.
>
> Our method performs significantly better than the fast ML-based rigid docking models DeepDock and UniMol, without even receiving the holo structure as input. Compared to large-scale co-folding models, it has similar RMSD performance but significantly better validity than Umol while being significantly faster (average of 11s per complex for FlexDock and 206s for Umol excluding its MSA computation step). AlphaFold3 performs significantly better, however, it has been trained on more than one order of magnitude extra data compared to FlexDock and Umol.
>
> The prediction quality (as measured by PB-valid checks) is competitive with AF3, and achieved in a fraction of AF3 inference time, ~12s (11 for docking, 0.3 for relaxation) on a single A100 80GB GPU vs 5 mins across 16 A100 GPUs for AF3.
>
> We hope the discussions and results address your questions! Please let us know if there are any further clarifications and opportunities to improve the score.

---

> > ### Comment · Reviewer_ZfZb · 2024-11-22
> >
> > Thank you for your responses, which address most of my initial concerns. I have a few additional questions and suggestions:
> >
> > - In Appendix Figures 7 and 8, a small subset of cases still shows relatively larger RMSD values between apo and holo conformations. While I appreciate the explanation that FlexDock identifies an optimal transport by shifting towards conformations with smaller deviations, I think it would be insightful to visualize the docking positions for both the ground truth and FlexDock’s predictions in a couple of these challenging cases (two examples would suffice). I'm interested to see the differences in the pockets between these cases.
> >
> > - There is an another paper [1] uses the two stage flow for posterior sampling. I'm also interested of its similarity and difference between unbalanced flow.
> >
> > [1] Ohayon, Guy, Tomer Michaeli, and Michael Elad. "Posterior-mean rectified flow: Towards minimum mse photo-realistic image restoration." arXiv preprint arXiv:2410.00418 (2024).

---

> > > ### Author Response · Authors · 2024-11-23
> > > **Response to Followup by Reviewer ZfZb**
> > >
> > > Thank you for your suggestions! Below are the responses to your follow-up questions:
> > >
> > > *Visualization of Large Conformational Changes*
> > >
> > > We have now added 2 figures for large conformational changes ($> 8A$) in Appendix F, Figure 11 e) and f).
> > >
> > > *Comparison to Posterior-Mean Rectified Flow*
> > >
> > > Thank you for the suggested reference. We were not aware of this work, and it was posted on arXiv after the ICLR submission deadline. We first summarize the work, and then compare similarities and differences to our unbalanced flow approach.
> > >
> > > **Summary**
> > >
> > > The paper focuses on the image restoration task -- Given an image X, and a corrupted version of X, denoted by Y, the goal is to provide a reconstruction of the clean image $\hat{X}$ under some perceptual distance constraint. A commonly adopted solution for this task is posterior sampling, where the reconstruction $\hat{X} \sim p_{X | Y}$ is shown to satisfy the perceptual distance constraint theoretically.
> > >
> > > However, this is not the most optimal solution (in terms of MSE). A previous paper [1] showed theoretically that the lowest MSE is achieved by first predicting the posterior mean $\hat{X}^*$, followed by computing the OT plan $T$ between  $p(\hat{X}^*)$ and $p(X)$, which is used to sample $\hat{X}$ using the pushforward $T(\hat{X}^*)$.
> > >
> > > The referenced work utilizes this theory to propose a new approach:
> > >
> > >  i) use a neural network to predict the posterior mean,
> > >
> > > ii) use the rectified flow (which often approximates the OT plan in practice), to approximate the pushforward $T(\cdot)$
> > >
> > > **Similarities to our work:**
> > >
> > > The high level motivations of both methods are similar, in trying to improve the approximation of the distribution transport. The rectified flow adopted for the second stage is analogous to our relaxation task, with similar intuitions for parameter choices (small $\sigma_s$ to only learn a projection from the posterior mean to the ground truth, vs small $c_{relax}$ to learn the corresponding projection from approximate to relaxed pose).
> > >
> > > **Differences to our work**:
> > >
> > > The referenced work uses the theoretical result derived in [1] to propose a two-stage approach based on predicting the posterior mean and using a rectified flow to refine it to a low MSE solution. In contrast, we focus on improving approximation error by relaxing the marginal preservation constraint in Flow Matching, with specific choice of couplings to make the flows explicitly smaller, and using a second model for reweighting samples. Further, we show that composing unbalanced flows is analogous to taking local gradient steps towards the desired final distribution. This composition naturally aligns with the manifold docking and relaxation steps for the flexible docking task.
> > >
> > > While the posterior mean prediction can be viewed as a reparameterization of the denoising score matching objective in the Euclidean space, this is not true for Riemannian manifolds (due to the corresponding $\exp$ and $\log$ maps. To our best knowledge, the theoretical result derived in [1] for posterior mean prediction thus would not provide an equivalent characterization for Riemannian manifolds for the denoising score matching objective.
> > >
> > > For flexible docking, rather than Euclidean space, it is more prudent to work with the corresponding relevant manifold degrees of freedom. Since the choice of coupling is flexible for UFM, we can accept / reject samples based on their distance (or RMSD) in Euclidean space, but allow the actual flow to be implemented over manifold degrees of freedom.
> > >
> > > [1] Dror Freirich, Tomer Michaeli, and Ron Meir. A theory of the Distortion-Perception Tradeoff in Wasserstein space. NeurIPS 2021
> > >
> > > We hope this addresses your questions, and are happy to provide any further clarifications!

---

> > > > ### Comment · Reviewer_ZfZb · 2024-11-23
> > > >
> > > > Thanks for the authors' explanation. It seems that when dealing with cases with larger RMSD, FlexDock would find another optimal relaxed pocket. It is interesting. I think this is a good paper and I've raised my score.

---

### Official Review · Reviewer_YA9F · 2024-11-02

**Soundness:** 3
**Presentation:** 3
**Contribution:** 3
**Rating:** 8
**Confidence:** 2

**Summary:**

This papers introduces Unbalanced Flow Matching (UFM), an extension of traditional flow matching, particularly tackling molecular docking problems considering protein flexibility. While traditional molecular docking methods assumed rigid structure for proteins limiting the accuracy, sometimes failing in scenarios including conformational changes of protein. Based on UFM, the authors propose the framework FLEXDOCK that considers flexible docking, and structure relaxation for optimized pose.

**Strengths:**

1. A new paradigm coined Unbalanced Flow Matching

The proposed objective is an extension of the traditional flow matching. Relaxing the strict marginal conservation allows more efficient and accurate mapping between distributions.

2. Two subtasks for flexible docking

By dividing flexible docking into two subtasks, manifold docking, and structure relaxation, the method results in more physically realistic structures.

3. Empirical results

FLEXDOCK outperforms prior methods on the PDBBind benchmark. Noticeably, this has been done using the same training data and model architecture as DiffDock-Pocket. Ablation studies also verify each component of the proposed method.

**Weaknesses:**

1. Flexibility and complexity tradeoff

While UFM has strength in flexibility compared to prior works, it inevitably seems to face more complexity than before. Could the authors briefly summarize the complexity compared to the algorithms in the prior works, if possible?

2. Choice of coupling

UFM seems to rely much on the coupling distribution. The authors have empirically chosen cutoffs, but this dependency may have problems for generalization. I may have misunderstood the part of choosing the coupling.

**Questions:**

1. UFM in extreme cases

Just curious, does UFM succeed even in extreme scenarios where the unbound (apo) and bound (holo) protein structures highly differ?

---

> ### Author Response · Authors · 2024-11-22
> **Response to reviewer YA9F**
>
> Thank you for the time dedicated to reviewing our work and the constructive feedback. Below we respond to each one of your questions or comments and integrate the clarifications in the updated version of the manuscript.
>
> *Flexibility and complexity tradeoff. Could the authors briefly summarize the complexity compared to the algorithms in the prior works, if possible?*
>
> The closest class of algorithms to UFM is represented by Flow Matching (FM). Compared to FM there are two key extra components that one could argue increase the complexity of the method:
> 1. the definition of the coupling distribution
> 2. a need for a discriminator model to reject some of the samples.
>
> (1) While the design space for a coupling distribution is large, in this work we use a very simple coupling distribution based exclusively on the definition of a similarity function and a cutoff. The former is often clear for the problem setting, the latter is a single hyperparameter that can be easily tuned.
>
> (2) Many generative modeling methods, especially in structural biology applications, already rely on confidence models [1,2] to select the best poses coming out of the generative model. Therefore, while the addition of a discriminator/confidence model does add some complexity, many methods already require this.
>
> *Choice of coupling. UFM seems to rely much on the coupling distribution. The authors have empirically chosen cutoffs, but this dependency may have problems for generalization. I may have misunderstood the part of choosing the coupling.*
>
> Indeed the choice of coupling plays a central role in UFM. However, we expect that for most scenarios, even simple choices of couplings are effective. For example, in our experiments, we focus on couplings dependent on a similarity metric (RMSD) and a cutoff.  Regarding generalization, we believe that UFM instead helps with generalization because it makes the learning task for the model easier. If one relies on normal FM, the model has to learn some very large conformational changes for a few proteins in the training set. The loss will be largely influenced by these examples, and the model will tend to just memorize them leading to poor generalization. Instead, the contribution to the loss in UFM is significantly more uniform across examples in the dataset potentially reducing the incentive for the model to overfit to any particular example.
>
> *UFM in extreme cases. Just curious, does UFM succeed even in extreme scenarios where the unbound (apo) and bound (holo) protein structures highly differ?*
>
> This situation is possible but somewhat rare because, while the distribution changes upon binding, the poses that the protein adopts when bound to the ligand are relatively close to poses that have good energetics even when unbound. However, when this happens during inference the model will try to find a good protein conformation for the ligand to bind nearby the apo structure. When only looking at the structures where the distance between apo and holo is greater than 4A, we observe that the distribution of generated poses by the model does tend to shift towards lower RMSD, but the shift is limited. However, the PoseBusters checks are still good on these poses (77.5% passing) suggesting that the poses are still physical. We added a full analysis on this in Appendix E.
>
> For such extreme scenarios, it could also be beneficial to have some diversity in the starting apo structures, rather than the single ESMFold structure we use currently. Current tooling for sampling the conformational landscape is limited, and any future improvements here could directly be combined with our UFM framework.
>
> We hope the discussions and results address your questions and concerns! Please let us know if there are any further opportunities for clarification and improving the score.
>
> [1] Corso, G., Stärk, H., Jing, B., Barzilay, R., & Jaakkola, T. (2022). DiffDock: Diffusion steps, twists, and turns for molecular docking.
>
> [2] Abramson, J., Adler, J., Dunger, J., Evans, R., Green, T., Pritzel, A., ... & Jumper, J. M. (2024). Accurate structure prediction of biomolecular interactions with AlphaFold 3.

---

> > ### Comment · Reviewer_YA9F · 2024-11-23
> >
> > I thank the authors for their detailed response and updated manuscript. I have raised my score accordingly.

---

### Official Review · Reviewer_R8VH · 2024-11-03

**Soundness:** 3
**Presentation:** 3
**Contribution:** 3
**Rating:** 8
**Confidence:** 3

**Summary:**

The authors propose FlexDock, a flow matching-based method for flexible small molecule docking. To address the problem of the limited availability of the bound protein structures, authors formulate Uncoupled Flow Matching, in which they tradeoff the approximation quality of the target distribution and the amount of samples needed from the prior distribution. Authors show that FlexDock outperforms other flexible docking methods in various metrics.

**Strengths:**

1. The proposed Unbalanced FM framework is a very neat idea and can be very useful in many different applications where the data from one of the distributions is limited
2. FlexDock demonstrates impressive performance across multiple metrics

**Weaknesses:**

1. In the motivation for the unbalanced flow matching authors say that due to the limited availability of the data points from $q_1$ (i.e. holo structures), the expected length of the flow will be large and the expected OT cost between $q_0$ and $q_1$ will be high. In my opinion, that would be true if there was a certain level of diversity in samples from $q_0$ (i.e. different apo structures) that correspond to a single (available) sample from $q_1$. But is it really the case for protein structures? I have a feeling that the diversity in apo structures of the same protein is not that high. I believe that the proposed method indeed improves learning the transport, as demonstrated in Figure 4. But I think it would be very beneficial to dissect this experiment and look at the flow distances and transport costs for different cutoffs, or even just plot these distributions for your training data. In my opinion, such a deeper analysis will strengthen the motivation a lot.

2. Authors suggest to use the coupling of the form $q(x0,x1) = q_0(x_0)q_1(x_1)\mathbb{I}_{c(x_0, x_1)<cdock}$.
* 2.1 Does it mean that some examples where conformation change upon binding is significant are completely discarded from the training?
* 2.2 If 2.1 is true, how does the model perform on similar examples (i.e. the ones that change a lot upon binding) from the test set?
* 2.3 Can it be true that "All-Atom RMSD% < 1A = 41.7%" means that around 40% of the test set are proteins that almost don't change upon binding, and the model (trained on couplings with rather strict $c_{dock}$) just doesn't change these structures meaning that it "recovers" holo conformations for these 40% correctly?


**Minor comments:**
1. Lines 40-45: could you please provide the references to the works you’re discussing here?
2. Are the protein states over the trajectory physically-realistic?
3. Could you please provide a more detailed description of y-axis in Figure 4 (include "% of passed predictions" or something like this)?
4. Line 273 - 274 -> I could not find the introduction of the coupling $q(x0, x1) = q(x0)q(x1)\mathbb{I}$ in the previous section. What are you referring to?
5. Line 182 – I am not sure if it is a typo or maybe I am wrong, but to me it looks like (3) is the lower bound of (2).

**Questions:**

1. It would be interesting to look at some examples of the model predictions (overlaid with the ground truth).
2. It would be helpful to provide the distribution of RMSD on $C_{\alpha}$ positions between apo and holo structures in the training and test data.
3. What is $q_0$ for ligands and how does the Unbalanced FM framework combine with the diffusion model for ligands?
4. What is the actual sampling efficiency of the final model? I.e. how many samples are required on average until the one exceeding the confidence threshold is obtained? Does the Runtime in Table 1 account for it? It would be interesting to look at the dependency of ESS on the selected $c_{dock}$ cutoff.
5. It has been recently shown that the conventional train/test splits of PDBbind (that authors use in this work), have a significant data leakage [1]. Is it possible that majority of 41% successful predictions (in Table 1) overlap with the training data?

[1] Corso G. et al. Deep confident steps to new pockets: Strategies for docking generalization //ArXiv. – 2024.

---

> ### Author Response · Authors · 2024-11-22
> **Response to reviewer R8VH 1/2**
>
> Thank you for the time dedicated to reviewing our work and the constructive feedback. Below we respond to each one of your questions or comments and integrate the new experiments and clarifications in the updated version of the manuscript.
>
> *Weaknesses*
>
> *1. I think it would be very beneficial to dissect this experiment and look at the flow distances and transport costs for different cutoffs, or even just plot these distributions for your training data. In my opinion, such a deeper analysis will strengthen the motivation a lot.*
>
> Thank you for the recommendation. We added this analysis to Appendix E. The analysis shows that while the distance between the pocket and the neighboring residues between apo and holo is largely within 4A, there are a number of examples beyond that and these significantly impact the expected squared coupling distance when using ordinary Flow Matching.
>
> *2. Authors suggest to use the coupling of the form q(x0,x1)=q0(x0)q1(x1)Ic(x0,x1)<cdock*
>
> Thanks for the questions on the coupling distribution. We've replied to each of them below and clarified these points in the manuscript.
>
> *2.1 Does it mean that some examples where conformation change upon binding is significant are completely discarded from the training?*
>
> Yes, if none of the samples from the apo distributions are close enough to samples from the holo distributions, the model will not train on those structures. Note however that for many proteins, while the distribution changes upon binding, the poses that the protein takes when bound to the ligand are relatively close to poses that have good energetics even when unbound.
>
> *2.2 If 2.1 is true, how does the model perform on similar examples (i.e. the ones that change a lot upon binding) from the test set?*
>
> In Appendix E, we plot the distribution of AA-RMSDs between apo and holo structures with large (> 4A) conformational changes, and their corresponding predicted and holo RMSDs. We find that FlexDock is able to shift the distribution towards lower RMSDs, albeit in a limited manner. This is also in intuitive agreement with the samples being transported to their closest coupling, based on the underlying (learned) transport map. Additionally, for these examples, we find that the generated structures are still physically realistic, validated by a 77.5% passing rate on PoseBusters checks.
>
> As an additional comment, for these extreme binding events, the inference would also benefit from having not just a single apo structure, but a few (ideally diverse) apo structures, so the transport with the learned vector field, and the corresponding reweighting with the confidence model can find structures that are close (in an RMSD sense) to the true holo-structure.
>
> *2.3 Can it be true that "All-Atom RMSD% < 1A = 41.7%" means that around 40% of the test set are proteins that almost don't change upon binding, and the model (trained on couplings with rather strict cdock) just doesn't change these structures meaning that it "recovers" holo conformations for these 40% correctly?*
>
> No, in fact the performance that one would get if none of the structures were changed from the original (i.e. using ESMFold) is 31.2%. This suggests that methods like DiffDock-Pocket (which obtains 32.4%) provide basically no noticeable improvement to the protein structure, while FlexDock does.
>
> *Minor comments:*
>
> *1. Lines 40-45: could you please provide the references to the works you’re discussing here?*
>
> Thanks for the recommendation we added some references to this component.
>
> *2. Are the protein states over the trajectory physically-realistic?*
>
> Not necessarily. The manifold flow guarantees that bond lengths and angles for the sidechains remain realistic. However, it is possible that the protein states (which are input to the model) have atoms from different sidechains clashing, making them non-physically realistic.
>
> *3. Could you please provide a more detailed description of y-axis in Figure 5 (include "% of passed predictions" or something like this)?*
>
> The oracle performance refers to the % of passed predictions when taking for every complex the best prediction across the different samples. We have improved the description in the caption.
>
> *4. Line 273 - 274 -> I could not find the introduction of the coupling q(x0,x1)=q(x0)q(x1)I in the previous section. What are you referring to?*
>
> Thanks for pointing this out, we have now fixed this reference adding a paragraph in the previous section.
>
> *5. Line 182 – I am not sure if it is a typo or maybe I am wrong, but to me it looks like (3) is the lower bound of (2).*
>
> You are correct. Thanks for pointing this out, we have now fixed this sentence.

---

> ### Author Response · Authors · 2024-11-22
> **Response to reviewer R8VH 2/2**
>
> *Questions:*
>
> *1. It would be interesting to look at some examples of the model predictions (overlaid with the ground truth).*
>
> We have now added a few example visualizations in Appendix F, with the ground truth and model predictions overlaid for the ligand and protein.
>
> *2. It would be helpful to provide the distribution of RMSD on Cα positions between apo and holo structures in the training and test data.*
>
> Thank you for the suggestion. We included a plot of the Apo-Holo AllAtom-RMSDs distribution in Appendix E. As can be seen by the plot most of the distribution lies below 5A, however, there is a very long tail of significantly larger distances.
>
> *3. What is q0 for ligands and how does the Unbalanced FM framework combine with the diffusion model for ligands?*
>
> q0 for ligands is the random prior as used by DiffDock: a ligand conformation obtained with a uniform random rotation, uniformly random torsion angles, and a large translation of the center of mass of the ligand. The way we combine Unbalanced FM and diffusion models in the manifold docking portion of FlexDock consists in modeling independently the forward diffusion\flow process i.e. the noise to the ligand is added independently of the protein conformation and the coupling of the protein structures is performed independently of the ligand. If a training example is rejected, the corresponding ligand is also unused (see also Algorithm 3 in the appendix). To model the joint distribution over protein and ligand poses, the architecture takes as input the structure of the full complex and predicts the denoising scores (for the ligand) as well as the flow vectors (for the proteins).
>
> *4. What is the actual sampling efficiency of the final model? I.e. how many samples are required on average until the one exceeding the confidence threshold is obtained? Does the Runtime in Table 1 account for it? It would be interesting to look at the dependency of ESS on the selected cdock cutoff.*
>
> In practice, rather than quantifying the actual sampling efficiency, we are often interested in implicit measurements of it, which can be used to compare two methods. For example, if method A is able to achieve better performance than method B while using a fewer number of samples, we say method A is more sample efficient. We have now included a figure for the same in Appendix E, plotting the docking performance of models for 1, 5, 10 samples, across different $c_\text{dock}$. We find that FM, even with 10 samples achieves worse docking performance than UFM with $c_\text{dock}=4.0$.
>
> In Table 1, the Runtime denotes the average time taken to generate 10 samples (a standard adopted across all baselines) per complex, and the performance is measured by the RMSD statistics of the top-ranked pose by the confidence model.
>
> *5. It has been recently shown that the conventional train/test splits of PDBbind (that authors use in this work), have a significant data leakage [1]. Is it possible that the majority of 41% successful predictions (in Table 1) overlap with the training data?*
>
> The conventional train/test splits of PDBBind indeed have similar proteins between train and test, however, the task of predicting the way that the protein conformation will change when bound to a new ligand is still challenging as demonstrated by the performance of the baselines (in-distribution generalization across complexes). While generalization to new protein pockets (out-of-distribution generalization to new complexes) is also important, we chose to use the conventional splits of PDBBind for two reasons:
> 1. Full protein flexibility (both backbone and sidechains) is largely unstudied in the ML docking community.
> 2. Understanding and improving protein flexibility modeling for the in-distribution setting thus presents a natural first step, and helps isolate performance improvements solely from modeling flexibility.
>
> The strategies proposed in the paper [1] for improving out-of-distribution generalization, namely model scaling, and self-training with the confidence model can also be applied to our method to potentially improve performance. This is however orthogonal to our contributions, and we feel is best left to future work.
>
> We hope the discussions and results address your questions and concerns! Please let us know if there are any further opportunities to improve the score.

---

> > ### Comment · Reviewer_R8VH · 2024-11-22
> > **Thank you**
> >
> > I thank the authors for the clarification and additional experiments. I raised my score.
> >
> > I additionally noticed a small typo in the RMSD formula in line 268.

---

### Official Review · Reviewer_7QfK · 2024-11-04

**Soundness:** 3
**Presentation:** 3
**Contribution:** 3
**Rating:** 6
**Confidence:** 3

**Summary:**

This paper extends Unbalanced Flow Matching (UFM), a novel approach to molecular docking that extends Flow Matching (FM) to better handle protein flexibility and generate physically realistic poses. The main innovation is allowing trade-offs between sample efficiency and approximation accuracy in the transport between distributions. The authors implement this in FLEXDOCK, which chains two UFM components: manifold docking for approximate pose prediction and structure relaxation for refinement. On the PDBBind benchmark, FLEXDOCK improves both docking accuracy and the proportion of energetically favorable poses compared to existing methods.

**Strengths:**

1. This paper attempts to address two critical limitations found in previous docking methods: the rigid assumption and the generation of non-physical structures.
2. The extensive experiments demonstrate the effectiveness of the proposed method. This paper demonstrates significant improvement in the PB-Valid metric, particularly with the simple yet effective energy loss.
3. The paper analyzes and formulates the flexible docking problem from the perspective of distribution transport, and extends the Flow Matching method specifically for this problem, supported by theoretical derivations.

**Weaknesses:**

The proposed method and experiments still assume prior knowledge of the binding pocket; despite this known pocket assumption, FLEXDOCK shows no significant improvement in RMSD-related metrics. This suggests that the manifold docking module may not make substantial progress in resolving the approximate pose compared to previous works.
However, this limitation appears to be inherent to the challenging nature of the protein-ligand docking problem and reflects the current state of the field, rather than diminishing the paper's existing contributions.

**Questions:**

1. Is the manifold docking module capable of blind docking, or is it limited to pocket-based docking only? (This is not a request for additional experiments, but rather an inquiry about the capability of this method.)
2. How sensitive is the method to the choice of hyperparameters?
3. What insights do the authors have regarding previous ML-based approaches performing poorly on the PB-Valid metric? Based on the experimental results from diffdock-pocket (rigid), rigid docking methods still generate non-physical structures. Which component predominantly drives the PB-Valid metric in your experiments - intramolecular or intermolecular validity?
4. Compared to PB-Valid metrics, FLEXDOCK demonstrates no significant improvement in RMSD-related metrics. While RMSD remains a crucial indicator for evaluating approximate poses, one must question the necessity of pursuing validity and physicality when RMSD values are still low and approximate pose estimation lacks precision.

---

> ### Author Response · Authors · 2024-11-22
> **Response to reviewer 7QFk 1/2**
>
> Thank you for the time dedicated to reviewing our paper and your feedback. Below, we respond to some of the main criticisms around the empirical results providing some needed clarifications and adjusting the manuscript accordingly.
>
> *The proposed method and experiments still assume prior knowledge of the binding pocket; despite this known pocket assumption, FLEXDOCK shows no significant improvement in RMSD-related metrics. This suggests that the manifold docking module may not make substantial progress in resolving the approximate pose compared to previous works. However, this limitation appears to be inherent to the challenging nature of the protein-ligand docking problem and reflects the current state of the field, rather than diminishing the paper's existing contributions.*
>
> Firstly, we would like to clarify that the other baselines provided in our paper also all have pocket conditioning information. This pocket conditioning information is indeed very important for the model's performance. For example, comparing the performance of our model with DiffDock, an established blind docking method that was trained on the same dataset, our performance in terms of ligand RMSD < 2A is 39.7% versus 21.7% of DiffDock (note that one has to take the performance when run on ESMFold protein structures, not holo protein structures).
>
> Regarding the RMSD metrics we would like to point out that we provide significant improvements over rigid methods and DiffDock-Pocket (which uses a diffusion model over sidechain conformations) when it comes to All-Atom RMSD < 1A. Our performance on AllAtom-RMSD is also somewhat better than the concurrent ReDock approach.
>
> Finally, while we agree with the reviewer on the challenging nature of the protein-ligand docking problem, we believe our empirical results, especially when it comes to the validity and physicality of the poses, provide a useful step towards a satisfactory solution to this problem.
>
> *Questions:*
>
> *1. Is the manifold docking module capable of blind docking, or is it limited to pocket-based docking only? (This is not a request for additional experiments, but rather an inquiry about the capability of this method.)*
>
> While the framework and the code can be readily expanded to perform flexible blind docking, we have not trained a blind docking model as part of this work.
>
> *2. How sensitive is the method to the choice of hyperparameters?*
>
> It largely depends on the specific hyperparameter in question. The most crucial hyperparameter in our framework is the selection of the cutoff $c_{dock}$ used to determine the coupling distribution. We provide a direct ablation of this hyperparameter in Figure 5, showing the importance of selecting an appropriate value. For the other hyperparameters, we largely directly adopted the ones provided by the baseline DiffDock-pocket and did not see very large performance variations from reasonable variations of these.
>
> *3. What insights do the authors have regarding previous ML-based approaches performing poorly on the PB-Valid metric? Based on the experimental results from diffdock-pocket (rigid), rigid docking methods still generate non-physical structures. Which component predominantly drives the PB-Valid metric in your experiments - intramolecular or intermolecular validity?*
>
> Previous ML-based approaches operating over manifolds (e.g. torsional and roto-translation degrees of freedom), tend to have good intramolecular validities but significantly struggle with the intermolecular interactions. For these approaches, corresponding local structures (e.g. bond lengths / bond angles) are held fixed. While holding them fixed is sufficient for obtaining approximate poses with good intramolecular validities, improvements to intermolecular validities require some modifications to the local structures, which is what we aim to capture with our fast (and short) relaxation. We have included a stratification of Posebusters validity checks in Appendix E.

---

> ### Author Response · Authors · 2024-11-22
> **Response to reviewer 7QFk 2/2**
>
> *4. Compared to PB-Valid metrics, FLEXDOCK demonstrates no significant improvement in RMSD-related metrics. While RMSD remains a crucial indicator for evaluating approximate poses, one must question the necessity of pursuing validity and physicality when RMSD values are still low and approximate pose estimation lacks precision.*
>
> See the comment above regarding RMSD comparisons. Regarding the question of whether one should already pursue validity, we argue so for two reasons: (1) the models are already accurate enough to be used in many important applications, thus having valid and physical poses is often very important for downstream analysis and prediction tasks, (2) we believe that the necessary components to solve the approximate pose prediction are largely orthogonal to those to solve the validity/relaxation challenge, therefore any contribution towards solving the latter problem will likely apply also to new models improving the first problem.
>
> We hope that our comments were useful to clarify the contribution of our work and its relation with the wider field, and address your concerns! Please let us know if there are any further opportunities for clarification and improving the score.

---

> > ### Comment · Reviewer_7QfK · 2024-11-23
> >
> > Thank you for addressing my concerns. While I still believe that docking faces other more significant bottlenecks, I acknowledge your point that solving approximate pose prediction is largely orthogonal to tackling the validity/relaxation challenge. I already consider this paper to exceed the usual standards for a paper at an ML conference, and I have raised my score accordingly.

---

### Official Review · Reviewer_sBsN · 2024-11-04

**Soundness:** 3
**Presentation:** 3
**Contribution:** 3
**Rating:** 8
**Confidence:** 3

**Summary:**

The paper presents a new approach named Flexdock for flexible molecular docking and structure relaxation.  The model is based on Unbalanced Flow Matching (UFM), which relaxes the marginal constraints of FM through a more careful chosen coupling distribution and reduces the modeling complexity. This framework aims to address limitations in existing molecular docking techniques, particularly concerning protein flexibility and the generation of nonphysical molecular poses. From the experimental results on PDBBind benchmark, the proposed method well achieved this goal, especially in generating energetically favorable poses.

**Strengths:**

* As far as I know, the introduction of unbalanced FM in flexible molecular docking is novel.
* The authors analyze sample efficiency and approximation error with mathematical foundations.
* The introduction of energy-based loss for relaxation is reasonable and seems very useful in generating energetically favorable poses.

**Weaknesses:**

* The introduction section lacks necessary citations. For example, in line 43, “Deep learning methods … often force the model to relearn protein folding”. What does “relearn” mean?  Please provide citations for such related work.
* In the related work section, the authors pointed out some related work but didn’t discuss the difference / advantages of their proposed method over related work.
* In methods section, the introduction of Eq3 and Eq4 is unclear. What’s their connection and how are they applied in practice?
* What is the influence of the choice of threshold c_{relax}?
* Have the authors tried using traditional (non learning based) energy relaxation methods like Rosetta as a step of post processing for baseline models and the basic FlexDock without proposed relaxation? I’m curious how it will perform compared to the proposed relaxation method.

**Questions:**

(Have already stated in the weakness section)

---

> ### Author Response · Authors · 2024-11-22
> **Response to reviewer sbSN**
>
> Thank you for the time dedicated to reviewing our work and the constructive feedback. Below we respond to each one of your questions or comments and integrate the new experiments and clarifications in the updated version of the manuscript.
>
> *The introduction section lacks necessary citations. For example, in line 43, “Deep learning methods … often force the model to relearn protein folding”. What does “relearn” mean? Please provide citations for such related work.*
>
> We have clarified the reference to relearning folding and added a few related work citations in the introduction section.
>
> *In the related work section, the authors pointed out some related work but didn’t discuss the difference / advantages of their proposed method over related work.*
>
> We have added these discussions to the related work section.
>
> *In methods section, the introduction of Eq3 and Eq4 is unclear. What’s their connection and how are they applied in practice?*
>
> Eq 3 describes a lower bound to the UFM objective, obtained by choosing the vector field v=0, which allows us to choose the coupling distribution $q$ without this choice being dependent on the parameters of the model that will be learned. Similarly, Eq 4 describes the optimization problem that results when minimizing the UFM objective under a particular choice of q, this is the loss function that is directly used during training. We have clarified these points in the paper now.
>
> *What is the influence of the choice of threshold $c_{relax}$?*
>
> Using a large $c_{relax}$ is equivalent to repeating the docking procedure. We want to have $c_{relax}$ small, so we learn modifications that only correct the steric clashes emerging from incompatible bond lengths and angles in the apo structure and ligand. Note that these local structures are treated fixed during the manifold docking step. Making $c_{relax}$ too small, however, can cause us to get stuck in local minima even when there are significantly better poses nearby. Empirically, we found $c_{relax}$ = 2A to be most effective.
>
> *Have the authors tried using traditional (non learning based) energy relaxation methods like Rosetta as a step of post processing for baseline models and the basic FlexDock without proposed relaxation? I’m curious how it will perform compared to the proposed relaxation method.*
>
> Thank you for the suggestion. We also conducted a relaxation using PoseBusters energy minimization tool (which relies on the OpenMM package and the addition of a number of constraints), with the results presented in the plots below:
>
> | Method                  | Time (System Prep + Relaxation) | PB-Valid (%) |
> |-------------------------|----------------------------------|--------------|
> | PoseBusters Minimization | 155s                           | 68.4%        |
> | Ours                    | 0.3s                           | 72.9%        |
>
> This highlights the efficiency and effectiveness of our relaxation framework. We added this discussion and comparison to Section 5.
>
> We hope the discussions and results address your questions and concerns! Please let us know if there are any further opportunities for clarification and improving the score.

---

> > ### Author Response · Authors · 2024-11-24
> > **Follow up on our response to Reviwer sbSN**
> >
> > Dear Reviewer,
> >
> > We once again thank you for your time and feedback.
> >
> > Since the rebuttal period is ending soon, we'd like to know if your concerns / questions have been addressed, and are happy to engage in any further opportunities for clarification and improving the score.

---

> > > ### Comment · Reviewer_sBsN · 2024-11-25
> > >
> > > I thank the authors for the response. My concerns have been well addressed. I'd like to raise my score to 8.

---

### Meta-Review · Area_Chair_5Wj1 · 2024-12-22

**Metareview:**

The paper proposes unbalanced flow matching to introduce a trade-off between the sampling efficiency from prior distribution and the error of the target distribution approximation. They optimize this trade-off for flexible docking with low holo structures and show improved results on standard docking datasets where they also achieve a higher number of low-energy and physical poses compared to baselines.

The reviewers acknowledged the justified direction of the work, novelty and generality of the technique and the significant improvement in the experimental results compared to recent works.

On the other hand, they were concerned about a proper coverage of related prior work, the choice of the hyperparameters, the choice of baselines, and whether the improvements actually come from the lower sample complexity.

The authors provided a rebuttal and revision with improved references, clarification regarding the sensitivity to hyperaparameters, provided new baselines with post-processing energy minimization of prior work, and conducted new experiments regarding the trade-off they introduce.

While the reviewers had diverging scores initially, they all suggest acceptance after the rebuttal which they mostly found satisfactory. The AC agrees with this unanimous assessment and suggests acceptance. The paper seems to be both of interest to the audience of molecular docking but also to the wider audience of flow matching generative models.

**Additional Comments On Reviewer Discussion:**

Five expert reviewers evaluated the paper and all were satisfied after the rebuttal.

---

### Decision · Program_Chairs · 2025-01-22

Accept (Oral)